# Polarization-driven reversible actuation in a photo-responsive polymer composite

David Urban ®[1,2], Niccolò Marcucci ®[2], Christoph Hubertus Wölfle[3], Jan Torgersen[3], Dag Roar Hjelme[1] & Emiliano Descrovi ®[2] ✉

Light-responsive polymers and especially amorphous azopolymers with intrinsic anisotropic and polarization-dependent deformation photo-response hold great promises for remotely controlled, tunable devices. However, dynamic control requires reversibility characteristics far beyond what is currently obtainable via plastic deformation of such polymers. Here, we embed azopolymer microparticles in a rubbery elastic matrix at high density. In the resulting composite, cumulative deformations are replaced by reversible shape switching – with two reversible degrees of freedom defined uniquely by the writing beam polarization. We quantify the locally induced strains, including small creeping losses, directly by means of a deformation tracking algorithm acting on microscope images of planar substrates. Further, we introduce free-standing 3D actuators able to smoothly undergo multiple configurational changes, including twisting, roll-in, grabbing-like actuation, and even continuous, pivot-less shape rotation, all dictated by a single wavelength laser beam with controlled polarization.

In the fast-growing domain of stimuli-responsive materials, photo-responsive polymers hold a prominent position thanks to several advantageous features, such as their intrinsically contactless control and the large variety of light sources and compounds available to tailor material responses[1]. Amorphous sidechain azopolymers are an intriguing sub-class of light-responsive polymers since many of them can undergo directional, anisotropic deformations when irradiated with linearly polarized visible light. This effect occurs below the glass transition temperature, with the polarization direction generally determining the axis of elongation[2]. Although this phenomenon is known to be linked to the Weigert effect[3] (statistical reorientation of azobenzene moieties perpendicular to the polarization), different mechanisms are still under debate to explain it[4,5]. In the meantime, polarization-dependent deformations in side-chain azopolymers have been widely used for the thin film inscription of surface relief gratings (SRGs) by pure polarization interference patterns[6,7], polarization-dependent patterning through surface plasmon interference[8], as well

as directional reshaping of both nanoparticles[9,10] and pre-fabricated micro-/nanopillar structures[11–15]. Whilst these approaches are very appealing to the realm of advanced micro- and nanofabrication techniques[16], or self-healing devices[17], deformations are usually plastic and tend to apply to the materials in a cumulative manner. Achieving deformation reversibility has therefore been a major goal in recent works, where approaches included the use of azopolymer cross-linked networks[18,19], erasure of SRGs by complementary interference patterns or circular polarization irradiation[20], and embedding of azopolymer microparticles in elastomeric materials[21]. Furthermore, in concrete applications such as photo-switchable topographies guiding cells, obtaining intrinsic reversibility has been identified as a key challenge[15,22,23].

Controlling the deformation direction and reversibility is also of paramount importance in light-responsive soft actuators. In this context, working schemes have involved linear actuators based on asymmetric volume expansion in various materials such as hydrogels or

[1]Department of Electronic Systems, Norwegian University of Science and Technology, O.S. Bragstads plass 2b, 7034 Trondheim, Norway. [2]Dipartimento di Scienza Applicata e Tecnologia, Politecnico di Torino, Corso Duca degli Abruzzi 24, 10129 Torino, Italy. [3]Institute of Materials Science, Department of Materials Engineering, TUM School of Engineering and Design, Technical University of Munich, Boltzmannstraße 15, 85748 Garching, Germany. ✉e-mail: emiliano.descrovi@polito.it

azopolymer-coated substrates[24,25], bidirectionally actuatable cross-linked azo-polyimides[26], and photo-responsive liquid crystal elastomers/networks (LCNs)[27]. The latter is maybe the most well-known type of light-responsive soft actuators[28] and can benefit from excellent reversibility and actuation speeds, due to their cross-linked nature, which permits the design of light-responsive oscillating[29–32], wave-propagating[33], and even mobile actuators[33–36]. However, LCNs usually require a molecular alignment procedure to be performed during fabrication, which, at least locally, fully predetermines the directionality of their response. In addition, inscribing the sophisticated alignment patterns needed for complex actuation shapes is still far from being trivial[37,38]. To overcome these limitations, several post-fabrication tunability strategies have been proposed. Examples include multi-wavelength schemes with several dyes[39–41], the use of constituents exhibiting an enhanced polarization-sensitivity in absorption[42], or reconfigurability based on a combination of photochemical and photothermal effects[43]. Polarization-driven actuation in polycrystalline LCN has also been proposed as an interesting option[44,45]. However, performing reversible complex actuations at room temperature via polarization-driven control remains a relevant challenge.

Here we introduce an isotropic composite based on amorphous azopolymer nano-/microparticles embedded at high density within a soft rubbery Styrene-Ethylene-Butylene-Styrene (SEBS) matrix, the latter having been previously used in electrically driven soft actuators[46]. The azopolymer employed is poly [(methyl methacrylate)-co-(Disperse Red 1 methacrylate)] (pDR1m-co-mma, characterization details provided in Supplementary Fig. 1), and the composite will simply be referred to as azo-SEBS in the following. Upon irradiation with linear polarization, the azopolymer particles transmit their anisotropic strain along the polarization direction to the overall matrix, thus producing a controlled deformation. During consecutive irradiation steps, cumulative deformations that would be expected from the uncross-linked amorphous azopolymer are replaced by a more reversible behaviour, which emerges from the interaction of the particles with the surrounding elastomeric matrix. A similar effect has recently been observed for sparsely distributed, single azopolymer microparticles[21].

In the following, a detailed analysis of the local in-plane deformation of thin composite layers of azo-SEBS deposited on polydimethylsiloxane (PDMS) slabs is presented, providing a benchmark of the overall degree of reversibility upon several illumination cycles. Thereafter, from an application perspective, soft actuators based on free-standing membranes are proposed. We showcase that ample,

reversible, and continuous complex actuation can be performed by merely controlling the polarization state of an illuminating laser beam. In Fig. 1, the fundamental actuation scheme for both thin coating layers and free-standing actuator membranes is illustrated.

## Results

### Deformation of planar substrates

To characterize the stretching of the azo-SEBS composite quantitatively, the thin coating layers (thickness 3-20 μm) were decorated with a periodic pattern of micro-pillar markers for precise deformation tracking (Fig. 2a). Since all light-induced deformations were observed to persist with switched-off illumination, the fast-scanning focused laser beam of a confocal microscope was used to illuminate tens of microns wide rectangular areas precisely and homogeneously. A movie of the repeated anisotropic deformations induced by a sequence of illumination steps with alternating linear and circular polarization of the scanning beam (wavelength $\lambda = 561$ nm, intensity $I = 5.02$ W·cm$^{-2}$, step time $t = 15$ s) is provided in Supplementary Movie 1. Whilst polarization along the x-axis induces stretching along said direction, circular polarization seems to induce a return to the previous state. The associated material displacements can also be observed to extend beyond the directly illuminated zone shown in the movie, to accommodate the deformation of the latter. This is displayed in Supplementary Fig. 2, where a differential image of a wider area was used to compare the initial state and the stretched state after a single illumination step with x-oriented polarization.

Deformations within the area of irradiation were then analyzed in Fourier space (Fig. 2b), where reciprocal lattice peaks of the hexagonal pattern of cylindrical markers are detected and tracked at each illumination step. This approach of focusing only on the light-induced changes in the spatial harmonics of the lattice was found to provide greater independence from small lateral and out-of-plane sample drifts, as well as tiny contrast changes produced in the collected images over many repeated illumination steps. The photo-responsive layer's adhesion to the underlying PDMS substrate also mitigates effects such as gradual out-of-plane bulging and wrinkling, which we observed in similar experiments on suspended azo-composite layers (Supplementary Movie 2, Supplementary Fig. 3) and which have recently been reported for floating azopolymer thin films[47]. In the Methods section, more details on how polarization-induced deformations affect the film topography in both direct and reciprocal (Fourier) space are provided. Red arrows in Fig. 2c depict the vectorial shifts $\Delta\mathbf{k}$ of the lattice peaks after an illumination step with linear x-axis polarization, taking the configuration before illumination as a

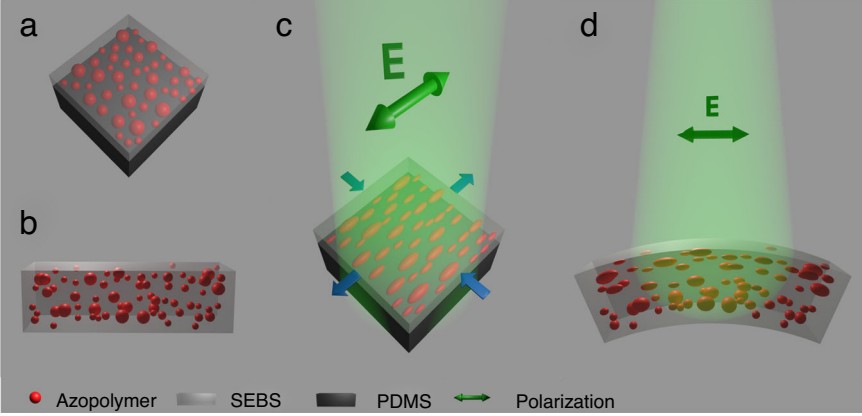

**Fig. 1 | Sketch of light-induced deformation mechanisms. a** azo-SEBS film deposited on a PDMS layer, (**b**) self-standing azo-SEBS membrane; (**c**) illustrative deformation of azopolymer microdomains upon linearly polarized laser irradiation, leading to overall stretching of the azo-SEBS layer on PDMS along the polarization direction; (**d**) inhomogeneous stretching of the free-standing azo-SEBS membrane and corresponding bending caused by the gradient of absorbed light through the membrane. Both SEBS and PDMS are translucent/transparent materials in the real world.

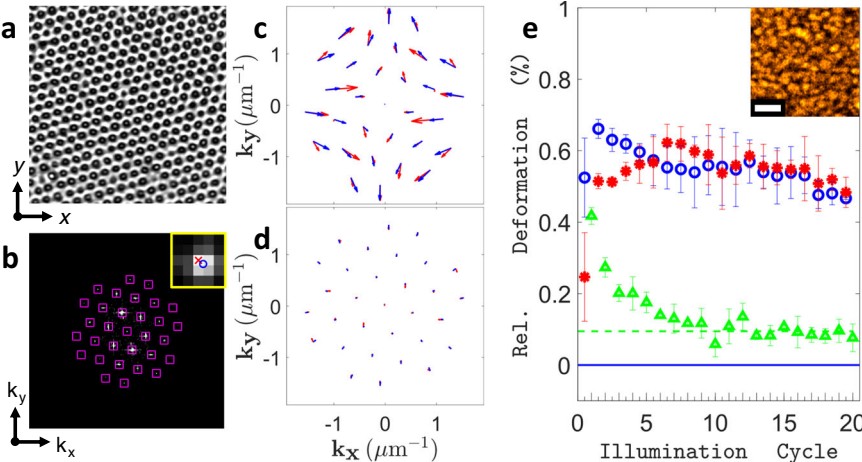

**Fig. 2 | Analysis of 2D deformations. a** Confocal microscope transmission image of the $33.8 \times 33.8\ \mu m^2$ area decorated with $1\ \mu m$ spaced, $1\ \mu m$ wide pillars, before illumination. **b** Fourier transforms the image of a, with detected peak locations (pink squares). Inset: 2D Gaussian fitting of exact peak position. Shift of peaks in Fourier space after irradiation with (**c**) linear polarization and after irradiation with (**d**) linear and subsequent circular polarization (illumination cycle). Red arrows: data, blue arrows: fit. Arrow magnification: x45. **e** Extracted fit parameters for 20 illumination cycles: relative horizontal elongation strain $\varepsilon_1$ per X-POL step (blue circles), absolute value of relative vertical compression strain $|\varepsilon_2| = -\varepsilon_2$ per X-POL step (red asterisks), relative area expansion $\delta A_{cycle}$ per full illumination cycle (X-POL + CIRC-POL) (green triangles). Dashed green line: asymptotic behaviour. Solid blue line: zero deformation limit. Short/tall x-axis ticks: X-pol step/two-step illumination cycle. Error bars: Sample standard deviation (s.d.) of 3 acquisitions on distinct areas. Inset: Confocal image of the sample's fine structure (bulk) showing the size of the azopolymer domains. Inset scale bar: $3\ \mu m$.

reference. In agreement with the x-axis elongation observed in direct space, peaks shift inwards along said axis in reciprocal space, whilst direct space compression behaviour leads to arrows pointing outwards in the orthogonal direction. Figure 2d shows a similar arrow plot, wherein the reference configuration is compared to the result after two irradiation steps, with a linear x-polarization (X-POL) step followed by a circular polarization (CIRC-POL) step (equal power and dose). This two-step procedure is referred to as the full illumination cycle hereafter. For the full illumination cycle, arrows are much smaller and point inwards everywhere, thereby indicating slight isotropic in-plane expansion, as opposed to complete reversal. In-plane expansion of azopolymers upon irradiation with circular polarization has been previously reported in microparticles dispersed in external matrices[21,48], microstructures anchored on flat substrates[12,17], wall thickening of breath-figure arrays[18,49] and in floating thin films[47]. It is often described to compete with a photo-softening effect of the azopolymer, also attributed to the circularly polarized irradiation[21,47]. To get more insights on the effects of repeated exposures with linearly and circularly polarized light, we quantified the stretching behaviour over 20 consecutive illumination cycles on a previously unexposed area.

In particular, when considering the $i^{th}$ illumination cycle, new experimental lattice peak positions were fitted with $k_i$ defined upon applying the transformation in Eq. (1) to the previous peak vectors $k_{i-1}$. The transformation represents pure stretching along two principal axes in Fourier space, with associated direct space engineering strain parameters ($\varepsilon_1$, $\varepsilon_2$) and a principal stretching axis angle ($\theta$) ranging between ±45 degrees with respect to the x-axis.

$$k_i = \begin{pmatrix} \cos(\theta) & -\sin(\theta) \\ \sin(\theta) & \cos(\theta) \end{pmatrix} \cdot \begin{pmatrix} (1+\varepsilon_1)^{-1} & 0 \\ 0 & (1+\varepsilon_2)^{-1} \end{pmatrix} \cdot \begin{pmatrix} \cos(\theta) & \sin(\theta) \\ -\sin(\theta) & \cos(\theta) \end{pmatrix} \cdot k_{i-1}$$

(1)

Furthermore, the stretching due to the illumination with linear polarization only was evaluated by applying the procedure described above to the peak positions obtained after the X-POL step alone (half illumination cycle), using the same reference positions $k_{i-1}$.

Typical $R2$-values of the fits (see Methods section) were $0.71 \pm 0.03$ for X-POL steps and $0.49 \pm 0.11$ for illumination cycles

respectively. The angle $\theta$ was distributed as $-4.3 \pm 1.4$ degrees for the X-POL steps, making $\varepsilon_1$ close to horizontal, as expected from linear x-polarization inducing stretching along said axis. For full illumination cycles, it was found more useful to estimate the relative area expansion (per cycle) of the substrate, which can be expressed as $\delta A_{cycle} = \varepsilon_1 + \varepsilon_2$ (first-order approximation). Therefore, in the following $\varepsilon_1$ and $\varepsilon_2$ will refer to the X-POL steps, while $\delta A_{cycle}$ refers to the illumination cycles (X-POL + CIRC-POL). The typical time evolution of $\varepsilon_1$, $|\varepsilon_2|$ and $\delta A_{cycle}$ for 20 consecutive full illumination cycles at laser intensity $I = 1.30\ W \cdot cm^{-2}$ and step time $t = 15\ s$, is shown in Fig. 2e.

Interestingly, one can observe an initial transient behaviour, where the horizontal elongation strain $\varepsilon_1$ upon linear x-polarized irradiation is much larger than the absolute value of the associated vertical compression strain $|\varepsilon_2| = -\varepsilon_2$, approximately by a factor 2. However, after a few cycles, the absolute values of the two strains converge to a very similar and nearly constant value. The asymptotic values for both strains will be referred to as $\varepsilon_{1,\infty}$ and $\varepsilon_{2,\infty}$ hereafter and are calculated as the respective mean values of $\varepsilon_1$ and $\varepsilon_2$ over the last 10 X-POL steps. Furthermore, similar transient behaviour can also be observed for the relative area expansion $\delta A_{cycle}$ at each illumination cycle. Starting at a high initial value, $\delta A_{cycle}$ decreases rapidly and stabilizes to a small and constant value $\delta A_{cycle,\infty}$, indicating much less in-plane area expansion and thus significantly better relative reversibility after a few cycles.

To explain this behaviour, we note that in a small strain regime, a factor $-\varepsilon_2 / \varepsilon_1 = 0.5$ is associated with volume conservation for uniaxial stretching, whilst $-\varepsilon_2 / \varepsilon_1 = 1$ conserves the area of observation. In the following, we presume that the observed macroscopic engineering strains are proportional to the deformations of the embedded azopolymer microdomains. This assumption is discussed with the help of a finite element (FE) model in the next section. In Eq. (2), we write down isochoric (volume-preserving) small strain deformation gradient tensors corresponding to x-axis stretching ($\mathbf{F^1}$) and in-plane expansion ($\mathbf{F^2}$) respectively. Both tensors are constrained to induce the same z-axis compression and $\delta$ is a scalar parameter.

$$\mathbf{F^1} = \begin{pmatrix} 1+\delta & 0 & 0 \\ 0 & 1-\tfrac{1}{2}\delta & 0 \\ 0 & 0 & 1-\tfrac{1}{2}\delta \end{pmatrix}, \mathbf{F^2} = \begin{pmatrix} 1+\tfrac{1}{4}\delta & 0 & 0 \\ 0 & 1+\tfrac{1}{4}\delta & 0 \\ 0 & 0 & 1-\tfrac{1}{2}\delta \end{pmatrix}$$

(2)

Assuming axial symmetry about the x-axis for the very first (X-POL) illumination step[50], the observed deformation seems to be associated with an isochoric stretching of the azo-microparticles along the x-axis. Indeed, a deformation described by $\mathbf{F^1}$ results in an engineering strain ratio $-\varepsilon_2/\varepsilon_1 \approx 0.5 = -(-1/2\delta/\delta)$. This agrees well with observations reported for free azopolymer nano-/micro-particles wherein irradiation with linear polarization isochorically transforms spherical particles into ellipsoids elongated along the polarization direction[10]. After completing the first illumination cycle with circular polarization, however, substantial in-plane expansion $\delta A_\text{cycle}$ subsists. Thus, in opposition to previous reports on single particles of a different side-chain azopolymer embedded in SEBS[21], circular polarization does not reverse the initial deformation here. In some cases, spherical azo-particles irradiated with circularly polarized light were reported to expand in the plane into disk shapes, as evidenced in previous works on azobenzene-based molecular glass particles in hydrogels[48]. The transformation corresponding to this deformation can be described by $\mathbf{F^2}$. Interestingly, however, after multiple illumination cycles, $\delta A_\text{cycle}$ is reaching a very low asymptotic value meaning that the in-plane area is almost fully conserved through each cycle. Still, within those cycles, the X-POL illumination steps cause relatively high, constant asymptotic stretching amplitudes $\varepsilon_{1,\infty}$ and $\varepsilon_{2,\infty}$, therefore indicating a reversible, stationary deformation behavior. This evidence suggests that deformations induced by alternating illumination with linear and circular polarizations cannot be simply described by sequential applications of $\mathbf{F^1}$ and $\mathbf{F^2}$, which would produce a constant and substantial in-plane expansion during every cycle. As a possible explanation, we propose a reversible switching regime occurring asymptotically, between two distinct shapes of the azo-microparticles, resembling the ellipsoids elongated along the x-axis and the in-plane expanded disk-like shapes introduced above. These shapes are found experimentally to have a similar in-plane area ($-\varepsilon_{2,\infty}/\varepsilon_{1,\infty} \approx 1$) and similar compression along the z-axis, assuming incompressibility (see Methods section). We may therefore represent the switching from disk-like to x-axis stretched ellipsoid by a third transformation $\mathbf{F^3}$, defined as

$$\mathbf{F^3} = \mathbf{F^1}\cdot(\mathbf{F^2})^{-1} = \begin{pmatrix} \frac{1+\delta}{1+\frac{1}{4}\delta} & 0 & 0 \\ 0 & \frac{1-\frac{1}{2}\delta}{1+\frac{1}{4}\delta} & 0 \\ 0 & 0 & \frac{1-\frac{1}{2}\delta}{1-\frac{1}{2}\delta} \end{pmatrix} \approx \begin{pmatrix} 1+\frac{3}{4}\delta & 0 & 0 \\ 0 & 1-\frac{3}{4}\delta & 0 \\ 0 & 0 & 1 \end{pmatrix} \quad (3)$$

where the final expression is based on a first-order approximation (see Supplementary Note 1), and $\mathbf{F^3}$ is consistent with the area conservation during X-POL steps observed at later cycles. Also, the small overshoot of $\varepsilon_1$ during the first illuminations may be explained by the x-axis strain being larger for a transformation from a spherical to the x-axis stretched ellipsoid shape ($\mathbf{F^1}$), than from a disk-like to the same ellipsoid shape in the switching regime ($\mathbf{F^3}$). Similar reasoning holds for the y-axis strains and the initial undershoot of $|\varepsilon_2|$.

Obtaining a reversible switching regime is remarkable since the plastic deformations of sidechain azopolymers are typically cumulative. For example, SRGs with square/tetragonal geometries have been demonstrated by simply adding a second interference pattern on top of an already inscribed one[7,22,51] and multi-SRG superposition can additively create up to 12-fold rotationally symmetric structures[52]. Cumulative deformation sequences have also been used to create complex 3D shapes from pristine hemispherical microstructures[12]. For example, applying a linear polarization to pre-deformed pillars made of similar side-chain azopolymer poly(Disperse Red 1 methacrylate) (pDR1m)[15] does not erase the first inscribed shape, but simply adds another deformation, flattening the individual pillars further (Supplementary Fig. 4). On a macroscopic level, applying the methodology of

this work to a pure pDR1m-co-mma film on PDMS, weaker, however, fully cumulative deformations were observed (Supplementary Movie 3, Supplementary Fig. 5). In that case, the relative area expansion remained high for all cycles and the $-\varepsilon_2/\varepsilon_1$ relationship for X-POL steps stayed close to 0.5 (as in $\mathbf{F^1}$), indicating no switching behaviour. During later frames, degradation of the pillar lattice could be observed also here (Supplementary Fig. 5b). Finally, it should be noted that the reversibility of the switching process reported for azo-SEBS, while being far from a purely cumulative effect, is not complete, since $\delta A_\text{cycle}$ approaches a non-zero value indicating that a (much smaller) irreversible in-plane expansion persists during later cycles also in the case of the composite.

## Finite element modeling of the deformation mechanism

In the previous section, the observed macroscopic engineering strains of the composite were assumed proportional to the deformations of the azopolymer microdomains embedded within. To verify the validity of this assumption, as well as to better understand the microstructural strain field causing the composite's macroscopic deformations, a 2D continuum model of a representative section of azo-SEBS composite with randomly placed, non-intersecting domains of pDR1m-co-mma (azo-domains) was set up using the FE software Abaqus/Standard. A detailed description of the model is provided in the Methods section, whilst further comments on the accuracy of the 2D modelling approximation are provided in Supplementary Note 2. In brief, both phases (azo-domains and SEBS) were considered incompressible and the constitutive model of the rubbery matrix was parametrized with mechanical data from tensile tests for SEBS of the same hardness grade and manufacturer performed by others[53]. As is common for rubbery materials, the stress-strain curve was shown to be non-linear at high strains in the cited study (see Supplementary Fig. 6a for more details). We note, however, that this non-linearity becomes dominant in such experiments at strains roughly above 20%, which is far above the macroscopic average strains measured in this work. To estimate the internal matrix strains associated with the composite's deformation, the effect of illumination was modelled via deformations imposed directly onto the embedded azo-domains. Linear x-polarization and circular polarization were modelled by the deformation gradient tensors $\mathbf{F^{lin}}$ and $\mathbf{F^{circ}}$, respectively, which are defined in the Methods section, and which correspond to $\mathbf{F^1}$ and $\mathbf{F^2}$ of Eq. (2) for small strains (first order approximation, see Supplementary Note 1).

Figure 3a shows the resulting logarithmic (Hencky) normal strain in x-direction, denoted by $\varepsilon_{xx}^H$, over the whole simulation domain. As one can observe, the highest strain values are the uniform ones imposed on the azo-domains, while the surrounding matrix accommodates the inclusions' deformation via lower local strains. Phenomenologically, compressive strains arise in x-direction between azo-domains lying close to each other horizontally, since the elongation of the inclusions in this direction compresses the material in between. Figure 3b shows the local relative area change (to the first order) $\varepsilon_{xx}^H + \varepsilon_{yy}^H \cong \varepsilon_{xx} + \varepsilon_{yy}$ also indicating a net in-plane compression between horizontally neighboring azo-domains. The opposite holds for vertically neighboring azo-domains, where in-plane expansion is observed, due to azo-domain compression along the y-axis. Both those observations illustrate how the deformation of individual domains drives the overall material's response. To expand on this, the coefficients of transmission between azo-domain deformations and the overall composite deformations will be discussed below along with their dependency on the magnitude of the imposed azo-domain strain.

After denoting the engineering strains imposed onto the azo-domains by $\mathbf{\varepsilon^{az}}$ and the average macroscopic engineering strains by $\mathbf{\varepsilon^m}$, we define the scalar transmission ratio of normal strains along any of the principal axes (i = x,y) generically as $T_{ii}^\text{type} := \varepsilon_{ii}^m / \varepsilon_{ii}^{az}$ (type = {lin, circ} when applying {$\mathbf{F^{lin}}, \mathbf{F^{circ}}$} to pristine spherical azo-domains respectively). Figure 3c shows $T_{ii}^\text{type}$ as a function of $\varepsilon_{ii}^{az}$ for both

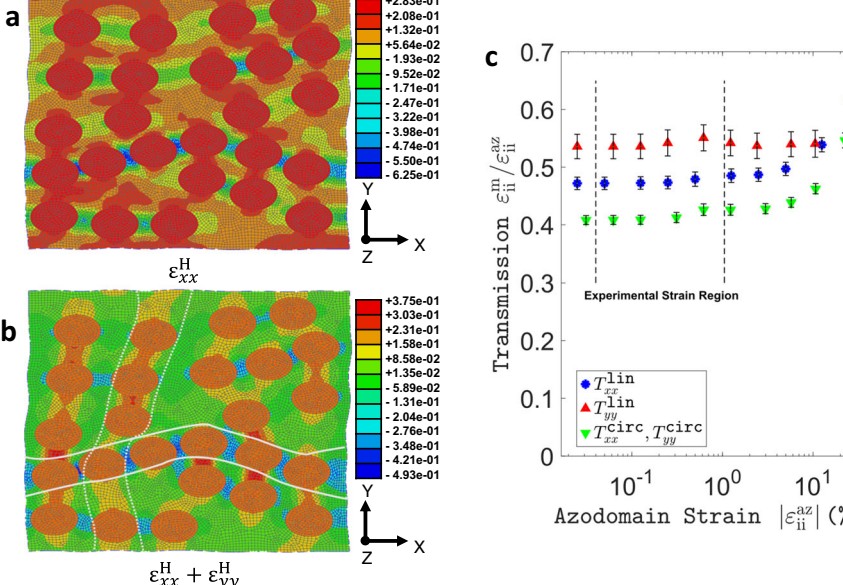

**Fig. 3 | Finite element model.** Logarithmic normal strain in x-direction $\varepsilon_{xx}^{H}$ (**a**) and first-order in-plane expansion $\varepsilon_{xx}^{H} + \varepsilon_{yy}^{H}$ (**b**) around the azo-domains stretched along the x-axis using $\mathbf{F}^{\mathrm{lin}}$ to model the illumination-induced eigenstrain of the azo-domains associated to linear x-polarization. Solid and dashed white lines provide a guide to the eye for identifying compression/expansion zones, respectively. The shear component and logarithmic normal strain in y-direction are shown in

Supplementary Fig. 6b, c. **c** Normal strain transmission factor $T_{ii}^{\mathrm{type}}$ as a function of imposed azo-domain strain $\varepsilon_{ii}^{\mathrm{az}}$ for each type of deformation (type = {lin, circ}) and principal axis (i = {x,y}). Error bars: sample standard deviation from 9 simulations with different randomly generated azo-domain arrangements. For simplicity of representation $T_{xx}^{\mathrm{circ}}$ and $T_{yy}^{\mathrm{circ}}$ are plotted with the same marker type due to their substantial overlap.

deformations studied and along either y- or x-axis, leading to four distinct data series. Since $\mathbf{F}^{\mathrm{circ}}$ has axial symmetry about the z-axis, the resulting values of $T_{xx}^{\mathrm{circ}}$ and $T_{yy}^{\mathrm{circ}}$ coincide. Further, one can observe that the transmission values do not depend on the amplitude of $\varepsilon_{ii}^{\mathrm{az}}$ in the range of experimentally observed strains. Finally, all transmission factors are rather similar, which was further verified for a sequential application of $\mathbf{F}^{\mathrm{lin}}$ and $\mathbf{F}^{\mathrm{circ}}$ deformations. Although slight offsets between the different data series, which are likely due to the 2D nature of the simulation (see Supplementary Note 2) can be appreciated, the results from the FE model support the qualitative assumption made in the previous section about linear transmission between the azopolymer microparticles' and the overall material's deformation.

**Varying illumination conditions and sample fine-structure**

The above FE model also predicted the transmission into macroscopic strains to be independent of microparticle deformation amplitudes. Therefore, we attempted to vary the latter by altering the illumination parameters. The result of varying laser powers and illumination step time is displayed in Fig. 4 for the asymptotic values $\varepsilon_{1,\infty}$ and $\varepsilon_{2,\infty}$. Both values are found to be dependent on the illumination step dose, rather than irradiation intensity, and follow the same sublinear trend. This sublinear behaviour agrees with reports on the deformation of single microparticles in a SEBS matrix as a function of irradiation time[21]. A similar trend is also observed for $\delta A_{\mathrm{cycle},\infty,}$ (see Supplementary Fig. 7) whose relative amplitude with respect to the stretching parameters is unaffected by the illumination parameters. Instead, this ratio can be sensitive to different fine structures of the azo-SEBS composite, as will be detailed below.

Note that all measurements up to this point have been carried out on what will be referred to as sample 1 hereafter. As described in the Methods section, it was possible to cast pDR1m-co-mma:SEBS blends at different degrees of dilution. In this way, samples with increasing thickness containing (larger) azopolymer aggregates could be obtained for decreasing amounts of solvent (Table 1).

Figure 5a shows the measurement of 20 actuation cycles conducted with the same illumination parameters which were used

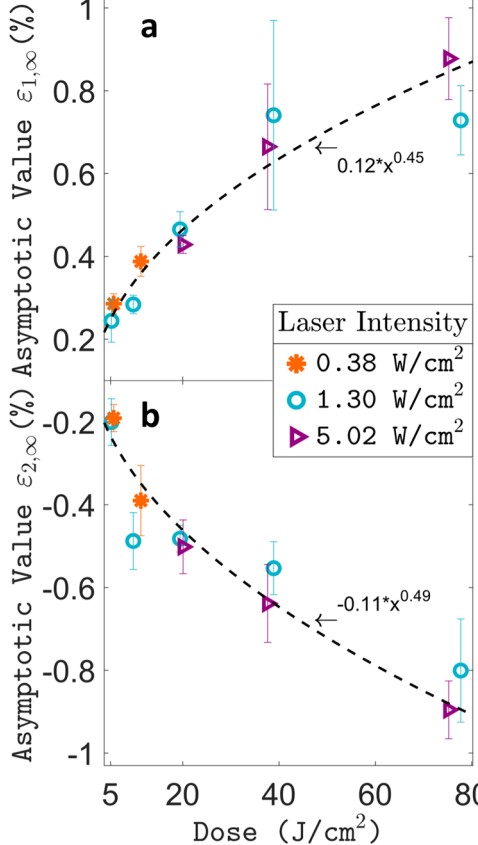

**Fig. 4 | Influence of illumination conditions on macroscopic strain.** Plot of asymptotic strain parameters $\varepsilon_{1,\infty}$ (**a**) and $\varepsilon_{2,\infty}$ (**b**) respectively as a function of applied dose (intensity times the exposure time) per illumination step. The asymptotic values are based on acquisitions whose graphs are shown in Supplementary Fig. 8. Error bars are three times the standard error of the mean (s.e.m.), as detailed in the Methods section.

**Table 1 | Summary of geometrical features and the asymptotic strain values obtained for different samples, made from different solid content during the casting process**

| Name: | Solute @ casting (wt%) | Estimated aggregate size (µm) | Estimated thickness (µm) | $\varepsilon_{1,\infty}$ (%) | $\varepsilon_{2,\infty}$ (%) | $\delta A_{cycle,\infty}$ (%) | $R_{irr}$ |
|---|---|---|---|---|---|---|---|
| Sample 1 | 0.7 | 1 | 3.1 ± 0.5 | 0.52 ± 0.03 | -0.54 ± 0.03 | 0.095 ± 0.004 | 0.18 ± 0.02 |
| Sample 2 | 5 | 2 | 5.4 ± 0.8 | 0.64 ± 0.07 | -0.45 ± 0.05 | 0.26 ± 0.02 | 0.41 ± 0.07 |
| Sample 3 | 10 | 15 | 12.9 ± 1.0 | 0.49 ± 0.06 | -0.40 ± 0.01 | 0.19 ± 0.03 | 0.40 ± 0.10 |
| Sample 4* | 15 | 75 | 14.4 ± 2.0 | - | - | - | - |
| *on top of aggregates | - | - | - | 0.10 ± 0.01 | -0.05 ± 0.01 | 0.080 ± 0.003 | 0.80 ± 0.09 |
| *between aggregates | - | - | - | 0.34 ± 0.07 | -0.33 ± 0.08 | 0.06 ± 001 | 0.18 ± 0.07 |

Uncertainties are based on the s.d. of three measurements for thickness, the s.e.m. of three independent acquisitions for $\varepsilon_{1,\infty}$, $\varepsilon_{2,\infty}$, $\delta A_{cycle,\infty}$ and on the error propagation for $R_{irr}$.

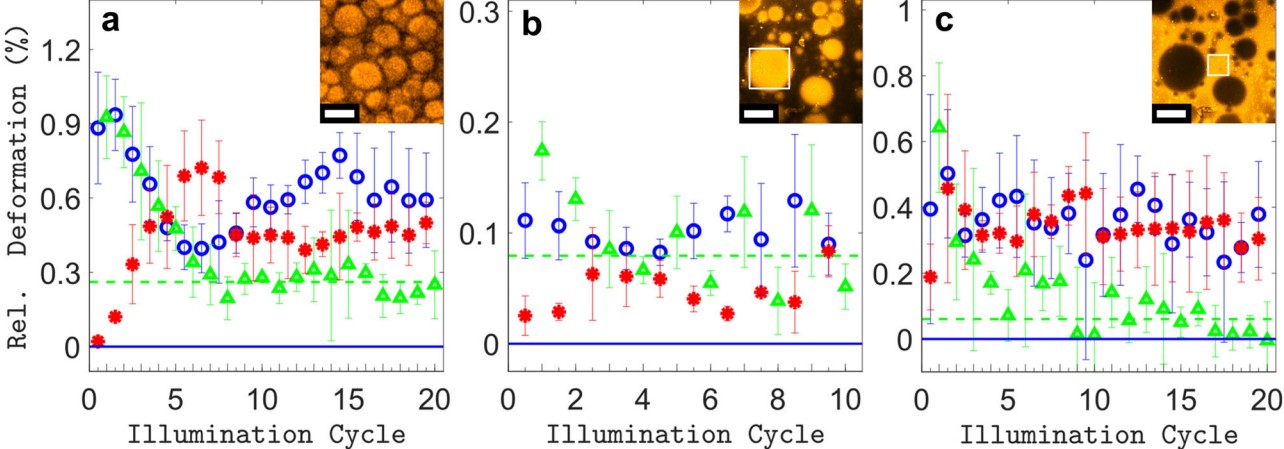

**Fig. 5 | Influence of sample fine structure.** Plot of fitted parameters for multiple actuation cycles for Sample 2 (**a**) and Sample 4 on top of large aggregates (**b**) and in between aggregates (**c**). Relative horizontal elongation strain $\varepsilon_1$ per X-POL step (blue circles), absolute value of relative vertical compression strain $|\varepsilon_2| = -\varepsilon_2$ per X-POL step (red asterisks), relative area expansion $\delta A_{cycle}$ per full illumination cycle (X-POL + CIRC-POL) (green triangles). Dashed green line: asymptotic behaviour. Solid blue line: zero deformation limit. Short/tall x-axis ticks: X-pol step/Two-step illumination cycle. Error bars: s.d. of 3 acquisitions on distinct areas. Insets: confocal images of samples' structures and aggregate sizes. Inset scale bars: (**a**) 3 µm, (**b**, **c**) 75 µm. White frames on insets in (b,c) show a typical illumination area.

previously (see e.g., Fig. 2e), yet carried out on sample 2, which has a slightly thicker azo-SEBS layer, with larger (<2 µm) azopolymer aggregates (see inset of Fig. 5a and Supplementary Fig. 9b). The most striking difference in this case is that the in-plane expansion per illumination cycle ($\delta A_{cycle}$) has an asymptotic value of 0.26 ± 0.02%, which is much higher than in the case of smaller azo-aggregates (sample 1). However, the anisotropic stretching values ($\varepsilon_{1,\infty}$, $\varepsilon_{2,\infty}$) remain similar. We therefore quantify the reversibility by defining an irreversibility factor $R_{irr} \equiv \delta A_{cycle,\infty}/\varepsilon_{1,\infty}$, a measure that will result in 0% for full reversibility in the switching regime and 100% for purely irreversible (cumulative) deformations. For sample 2, we obtained $R_{irr} = 41 \pm 7\%$ which, compared to $R_{irr} = 18 \pm 2\%$ for sample 1, highlights the higher irreversibility of this sample. In addition, $\varepsilon_{1,\infty}$ lies slightly above $\varepsilon_{2,\infty}$ for sample 2, confirming that more significant incompressible plastic deformation is occurring in the asymptotic regime, which for the case of uniaxial stretching along the x-axis indeed implies $\varepsilon_1$ higher than $|\varepsilon_2|$ (see $\mathbf{F^1}$ in Eq. (2)). To evaluate whether higher irreversibility is correlated to large azo-aggregates, sample 4, cast from even less solvent was considered. The largest aggregates were wide enough (<70 µm) to perform measurements in areas lying completely above a single aggregate. Results for 10 illumination cycles are displayed in Fig. 5b. In this case, deformations were much smaller, approaching the limit of the detection technique used. Nevertheless, the previously observed trend continued, with the irreversibility parameter reaching $R_{irr} = 80 \pm 9\%$, and the ratio $-\varepsilon_{1,\infty}/\varepsilon_{2,\infty}$ equaling 1.9, indicating almost fully plastic cumulative deformations at every illumination step (note the similarity to the graph obtained using a pure azopolymer film in

Supplementary Fig. 5a mentioned above). Measurements performed on the same sample, but on areas lying in between the large azopolymer aggregates (Fig. 5c) revealed an opposite behaviour with an irreversibility ratio as low as $R_{irr} = 17 \pm 7\%$, most likely due to the absence of big azopolymer aggregates in those interstitial areas (Supplementary Fig. 10). This suggests that not only the presence of larger aggregates but also their density could play a role in determining the average irreversibility of the overall material. In particular, interstitial volumes containing only smaller aggregates may continue to behave in a more reversible way. In fact, sample 3, which contains larger aggregates than sample 2, displays a similar average reversibility likely owing to this effect (Supplementary Fig. 11). The above findings are summarized in Table 1.

In conclusion, the preceding section shows that larger azopolymer aggregates should be avoided whenever possible, as they lose the shape-switching property and undergo the cumulative deformations of pure azopolymer instead. Furthermore, the analysis of illumination conditions shows that emerging shape-switching behaviour does not depend on a particular laser intensity or illumination dose within the investigated range. The dose however defines the amplitude of the induced deformations, meaning that intensity can be traded against illumination time, allowing for application-dependent adjustments.

**3D actuation of free membranes**
A potential application for the polymer composite presented herein is soft actuators able to perform complex movements in 3D[21]. To this aim, free-standing membranes were cut out and detached from PDMS,

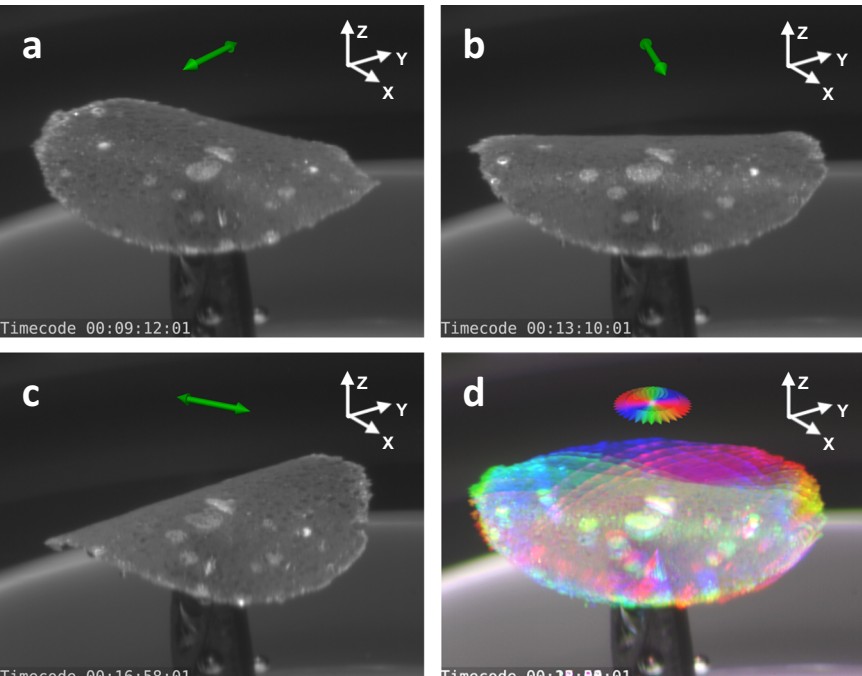

**Fig. 6 | Continuous rotation of bending axis on a circular membrane patch.**
**a**–**c** Snapshots from Supplementary Movie 4 showing different bent states in response to illumination with linear polarization in different directions (green arrow). **d** Rainbow-coloured chronological superposition of 18 movie frames, illustrating the continuous rotation behaviour. Illumination is provided from the top, along the z-axis.

in view of exploiting bending motion due to deformation gradients (Fig. 1b, d). Supplementary Movie 4 shows a circular membrane patch (diameter: 3.4 mm, thickness: ~20 μm) on the tip of a needle, experiencing downward bending along the continuously rotating axis of linear polarization of an incoming laser beam (top illumination). Selected frames from Supplementary Movie 4 are displayed in Fig. 6a–c and a superposition of 18 rainbow-coloured frames is shown in Fig. 6d. During the photo-actuation of the free membrane, deformations are observed to occur in a basically non-cumulative way as well. When the polarization orientation is altered, the induced expansion/compression of the membrane is found to simultaneously erase (most of) the previous deformation state, thus permitting a continuous shape rotation. Worth to mention in this context that the reversible switching behaviour on patterned planar films, introduced previously by analyzing (X-POL/CIRC-POL) illumination cycles, can also be observed in the case of alternating linear polarizations along the x-and y-axis (see Supplementary Fig. 12).

It should be noted that this continuous rotation was performed with the membrane immersed in water. In fact, as shown in Supplementary Movie 5, when applying the same illumination sequence with alternating perpendicular polarizations to a circular membrane in water and in air, a drastic difference in behaviour is observed. Whilst the membrane in water reversibly bends along the orthogonal polarization directions, without visible fatigue (7 steps), the first deformation shape in air seems irreversibly inscribed and subsequent illuminations lead to much smaller deviations from said shape. Furthermore, re-immersing the same membrane in water, the reversible behavior is not fully restored. Possible reasons for this behaviour are elaborated in the Discussion section.

Finally, the circular membrane was seen to mainly experience uniaxial, elongation-associated downwards-bending, away from the light source and in the direction of the illuminating polarization. Although not systematically investigated in this article, a tendency for smaller and thinner membranes to display more simultaneous upward bending in the direction orthogonal to the polarization (where compression was measured on planar substrates) can be reported. The

resulting saddle-shape, together with a single linear boundary condition, is exploited in the actuator experiment shown in Fig. 7. Here, a slightly trapezoidal strip of smaller dimensions (lateral: 1–1.2 mm × 1.6 mm, thickness ~10 μm) is glued to a glass slide, which forms a rigid boundary condition on the smaller side of the trapezoid (Fig. 7a). A colouring post-process is used to enhance the visibility of the actuator deformations (Fig. 7b). When linearly polarized light hits the sample from the top, elongation of the top surface in direction of the polarization causes downward bending, whilst the orthogonal direction shows compression-associated upwards-bending, leading to combined shapes. Merging the latter with a rigid boundary condition will produce both deformation and movement of the strip surface. For example, as shown in Fig. 7c, d, polarization at a 45-degree angle with respect to the constrained boundary causes one corner to bend upwards and one corner to bend downwards with respect to the boundary constraint, effectively producing a twist. Such photo-actuated polymer film twists have also been reported in the context of polydomain LCNs[45], cross-linked azo-polyimide cantilevers[54], and monodomain LCN strips with liquid crystal directors oriented diagonal to the cantilever[32,55]. In Ref. 55, repeated helicoidal (de-)curling of rectangular strips along a predefined direction was demonstrated. In the example presented herein, ample twisting motions to either side can be forced onto the soft strip by simply changing the polarization of the incoming light, permitting effective placement and reorientation of the actuator surface in 3D. Another interesting interplay between the two orthogonal bending directions and the boundary constraint is observed when the polarization is oriented parallel/perpendicular to the latter (Fig. 7e, f). Specifically, in Fig. 7e, the actuator front edge moves upwards whilst the two adjacent corners bend downwards simultaneously, resembling a grabbing motion. Switching the polarization by 90 degrees, the bending directions are reversed, and the tip now moves downwards, with the two adjacent corners bending upwards. It is worth noting that, in practice, the balance between the two orthogonal bending directions depends on a multitude of parameters, such as the geometry of the actuator, but also the amplitude of deformation. For example, strong deflections for downward-bending

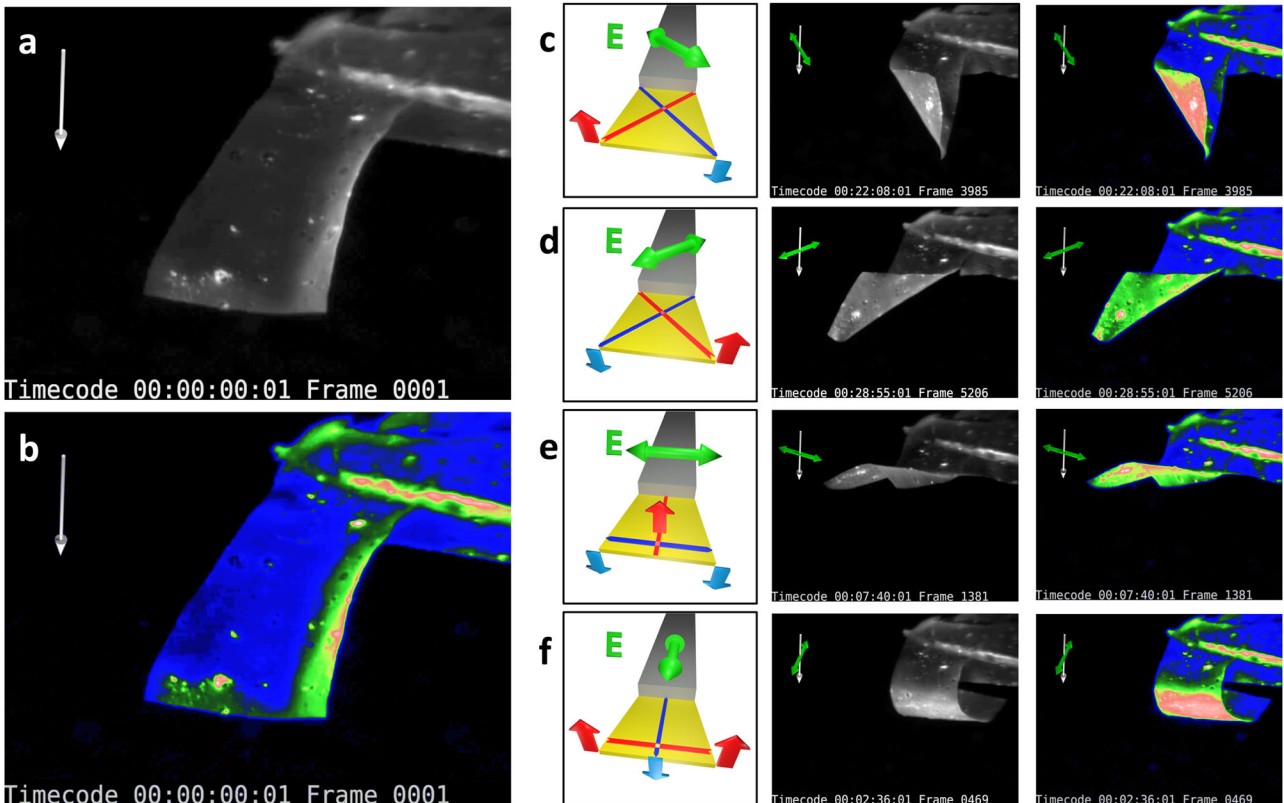

**Fig. 7 | Trapezoidal actuator. a** Snapshot of undeformed actuator. **b** False-coloured version of (**a**), highlighting contrast differences. **c**–**f** Image groups showing selected movie frames displaying twist to the right (**c**), twist to the left (**d**), upwards movement with transverse downward bending (**e**) and downwards movement with transverse upwards bending (**f**). Each image group consists of a schematic drawing of the deformation mechanism (left), one B/W image of the strongly deformed actuator (middle) and its false-coloured version (right). Green/white arrows: Polarization orientation/beam incidence direction.

were able to cause full roll-in of the strip, with weaker orthogonal bending (Fig. 7f). More generally, Supplementary Movie 6 allows a better understanding of all the specific features occurring during the full deformation sequence. In case circular polarization is employed, a general in-plane expansion of the illuminated side is observed (see Supplementary Fig. 13). A dependency of the strip movement on the previous deformations is observed at large actuation amplitudes, which can be attributed to the change of incidence angle of the laser beam onto the sample in motion. This may cause different projections of the polarization direction onto the sample surface, as well as sha-dowing effects and even the bottom surface of the sample being directly illuminated. Such effects have already been exploited to drive uniaxial oscillation of LCN strips with steady light stimuli[29,30,42,56]. To summarize, whilst the detailed actuation behaviour is more complex than one might expect at first glance, the overall nature of the deformation only depends on the polarization state. Moreover, fine-tuned experimental paradigms may lead to exploiting the more intricate details of the deformation pathways deliberately, for example, to design non-reciprocal movement sequences with single wavelength, homogeneous illumination[56].

## Discussion

The emergence of a reversible shape-switching behaviour upon embedding an amorphous azopolymer into an external rubbery matrix has been shown and quantified experimentally. Since both materials are commercially available, and no polymerization and/or molecule alignment techniques are needed, we believe this type of material will open new possibilities for wider use of polarization-sensitive light-responsive materials. As to why the switching behaviour emerges, several possibilities could be imagined: in the framework of the photo-orientation model, describing optically induced ordering of main chains parallel to the polarization as deformation driving force[47,50,57], one could assume that the deformation states saturate for the micro-particles, i.e., that the azo-domains' z-axis compression stops, when all main chains are oriented into the plane. However, the timescale for this (<1 min in all experiments) would be rather short, compared to other works[10,21]. In addition, longer illumination was shown to induce larger deformations at all stages, speaking against a saturation effect and rather for a continuous strain build-up[10], which however must be influenced by the previous state of deformation. One may therefore conclude that the switching behaviour is due to interaction effects between the azo-domains and the elastic SEBS matrix (e.g., surface tension, elastic restoring forces). Note that although the glass transi-tion temperature ($T_g = 108\,°C$) is not reached during laser irradiation (Supplementary Fig. 14), photo-softening effects may be present.

On the other hand, a small irreversible component of plastic deformation was measured also in the switching regime and shown to locally depend on the size of embedded azopolymer aggregates. A possible explanation lies in larger aggregates inducing higher local heating, relaxing the elastic constraints in the surrounding matrix. Indeed, SEBS is a thermoplastic elastomer, which is physically, not chemically, cross-linked, and whose thermoplastic creeping behaviour is expected to increase with rising temperature, up to the point where it permits injection molding of the material at temperatures above 150 °C. Another observation of the negative effect of heating on reversible actuation is the drastic improvement of reversibility repor-ted for 3D actuators immersed in water acting as a coolant, well known to play a crucial role for example in photo-thermal LCN actuators[35,58]. In this case, however, the increase of temperature is also due to a larger membrane thickness (see Supplementary Fig. 14), as compared to the

patterned thin films which showed the best reversibility. Finally, irreversibility stemming from other sources, such as aggregates interacting differently with the external matrix due to e.g., absorption gradients or larger volume/interface ratios, cannot be excluded. Future development of this type of material in view of beating the reversibility benchmarks set up in this work will likely include further reduction of large azopolymer aggregates.

We would like to point out that reversibility, as it was defined here, only refers to the continuous transition between in-plane expanded states and linearly stretched states along arbitrary directions, while the pristine state of the composite cannot be restored. Nevertheless, this leads to two reversible degrees of freedom, linked to the direction and the ellipticity of the polarization. In order to recover the very initial microdomain state, mechanisms based on heating the azopolymer above the glass transition temperature have been proposed and experimentally validated[21].

Regarding biological applications, the possibility of tuning both strain magnitude and direction locally may open for interesting experimental paradigms combined with deformations propagating readily outside of the directly irradiated area. Worth mentioning here is that a similar approach has been utilized previously in our group to modify topographic features around cells without directly exposing them to light, which could potentially affect their viability. In this case, however, cells had to migrate before retrieving the topographic cues[15].

Finally, the proposed 3D soft actuators showed ample bending deformations based solely on the absorption and thus strain gradient throughout the membranes' thickness, without a bilayer system to amplify bending[25,59]. Nevertheless, such an approach could become interesting in view of miniaturizing the system, which could lead to several application prospects. For example, the possibility of controlling smooth, continuous membrane deformations, generally not possible in polycrystalline photo-chemical LCN actuation where UV-inscribed shapes first need to be erased by visible light[44,60], may constitute an efficient tool to transfer rotation from a polarizer to the shape of microscopic membranes. In addition, the use of saddle-shapes, which although observed in monodomain LCNs[34], are usually not reported for polydomain LCN membranes[44,61], combined with boundary constraints, was shown to permit the precise inscription of complex membrane shapes. Such shapes may be particularly adapted to manipulating small objects, an application attracting increasing interest[58,62,63].

## Methods

### Photo-responsive layer preparation
Poly[(methyl methacrylate)-co-(Disperse Red 1 methacrylate)] (Sigma-Aldrich) was dissolved in toluene at 0.2 wt% whilst stirring for 1 h at 60 °C on a regular hotplate. Subsequently, block copolymer poly(styrene-ethylene-butylene-styrene) (SEBS, Mediprene 500120 M, Hexpol A/B) was added at 0.4 wt% and stirring at 60 °C was continued for at least 2 h. The solution was then cast on a micro hole-patterned polydimethylsiloxane (PDMS, Sylgard 184, DOW Corning) mold, either directly or after further evaporation of the solvent in an open beaker. The PDMS mold was obtained by soft lithography from an etched silicon master mold and treated with trichloro(1H, 1H, 2H, 2H-perfluorooctyl) silane (Sigma Aldrich) in a vacuum desiccator to avoid excessive sticking. Densified solutions (>1 wt% solid content) were directly sandwiched between the mold and an oxygen plasma treated (100% O2, 3 min, Diener Electronics Femto) flat PDMS receptive layer, then dried at room temperature, whilst the most dilute solution was transferred from the mold onto the receptive PDMS layer after drying (firm pressing, then bonding at 90 °C for 1.5 h). Free-standing membranes for 3D actuation were prepared via the same preparation protocol, but using two stiffer, fluorosilane-treated, flat PDMS (Sylgard 182, Dow) layers instead of a mold and a receptive layer. 3D actuators were then cut by means of a razor blade (trapezoidal membranes) or with a sharpened copper straw (round membranes). After they had been cut on the supporting PDMS, the membranes were carefully peeled off. To favour the peeling process and prevent self-wrapping caused by electrostatic forces, a few drops of deionized water mixed with salt (NaCl) and soap were poured on the surface of the membrane prior to peeling. Peeled membranes were laid onto a piece of oil-treated paper and spread out to remove all wrinkles, with the help of additional water droplets if required, before drying with optical grade cloth and/or low-temperature heat. The trapezoid membrane was glued to the edge of a glass slide with common fingernail varnish.

### Planar substrate actuation
The planar samples were imaged and actuated under a confocal microscope (Zeiss 800 Airyscan, Objective: Plan-Achromat 63x oil, NA = 1.4) using the 561 nm laser line for continuous scanning illumination (writing mode, default: linear horizontal polarization) and the 640 nm line for transmission or scattered laser beam imaging (reading mode) of the sample. For actuation with circular polarization, a $\lambda/4$ plate was inserted in the optical path via the beam-blocker slit and circular polarization of the laser beam was verified in polarized optical microscopy (POM) mode. The waveplate was removed for all image acquisitions. Irradiation powers for planar substrates were defined as laser power divided by the total rectangular scanning/illumination area ($33.8 \times 33.8$ – $101.4 \times 101.4$ μm$^2$) and in writing mode the scanning speed was set to its maximal value (pixel dwell time 0.59–0.91 μs, frame time 233–360 ms). Time-delay between the end of a writing illumination step and onset of the next writing interval was approximately 45 s.

### Planar substrates data analysis
In Supplementary Figure 15a, b, a schematic illustration of the deformations produced on a hexagonal pattern illuminated by a linearly polarized radiation is presented. For sake of generality, the incident linear polarization is oriented at an angle θ with respect to the x-axis. Two engineering strains $\varepsilon_1$ and $\varepsilon_2$ along two orthogonal axes are associated to light-induced stretching and compression, respectively. Such a deformation can be also described in the Fourier space as a set of transformations applied to the (reciprocal) hexagonal cell. As shown in Supplementary Fig. 15c–f, the initial lattice cell undergoes a first rotation by an angle – θ, then a stretching/compression along vertical and horizontal directions respectively, and finally a second rotation by an angle θ. Worth recalling that stretching in the direct space corresponds to compression in the reciprocal space and vice versa. Such transformations in the Fourier space are embodied in the matrix relationship of Eq. (1) used for fitting experimental data.

Data Analysis on collected images was conducted in MATLAB 2021®. After a 2D fast Fourier transform, peaks with a prominence above a peak prominence threshold were identified using the find-peaks function and further selected based on having the largest prominence in a neighborhood sized slightly below the expected pillar marker reciprocal lattice constant. The position was further adjusted using a two-dimensional Gaussian fitting procedure on the 25 pixels surrounding the detected peak. Finally, peak shifts between images were found by comparing the respective peak sets and forming pairs of peaks if their distance is below a limiting distance. They were displayed using MATLAB's quiver plot feature and fitted with a least squares approach using Eq. (1), to extract the engineering strain parameters ($\varepsilon_1$, $\varepsilon_2$) and the angle θ between the axis of $\varepsilon_1$ and the x-axis. The code is fully automatized and made available.

The $R2$ value of the reciprocal space fitting results was defined in analogy with the linear case as

$$R2 := 1 - \frac{\sum_i ||\mathbf{a}_{f,i} - \mathbf{a}_{d,i}||_2}{\sum_i ||\mathbf{a}_{d,i}||_2} \tag{4}$$

where $\mathbf{a}_{f,i}$ and $\mathbf{a}_{d,i}$ denote the fitted and the data arrows respectively (Fig. 2c, d). Where parameters are given as a single value for a whole actuation cycle set ($R2$, $\theta$), their value and spread are provided as mean and sample standard deviation (s.d.) for the whole actuation set. The s.d. was also used to compute error bars on Fig. 3c, based on 9 randomly initialized simulations, and to plot error bars for $\varepsilon_1$, $\varepsilon_2$, and $\delta A_{\text{cycle}}$ based on 3 acquisitions in different areas for the illumination sequences displayed in Fig. 2e and Fig. 5. The asymptotic values $\varepsilon_{1,\infty}$, $\varepsilon_{2,\infty}$, $\delta A_{\text{cycle},\infty}$ (Table 1) were defined as the mean of the last 10 values of the respective parameters in such averaged illumination sequences, 8 values in the case of Fig. 5b. Their uncertainties were computed as the standard error of the mean (s.e.m.) over the 3 individual 10-point means. The spread (s.d.) of those individual 10-point means was found on average over all measurement series to be ~3 times larger than the spread expected by computing the s.e.m. over the last 10 points for individual acquisitions, indicating that new acquisitions in different areas contribute in a non-negligible manner to the spread of the mean. Therefore, for the dose plot in Fig. 4, who relies on single acquisitions, measurements were carried out in closely neighboring areas and error bars estimating the spread were computed as the s.e.m. of the last 10 points corrected by a multiplicative factor 3.

## Planar substrate further characterization

The thickness of the active membranes was estimated by cross-section imaging with a tabletop scanning electron microscope (SEM) after sputter-coating of 10 nm of Pt-Pd. Example SEM images are shown in Supplementary Fig. 9a. Thicknesses and uncertainties provided in Table 1 are based on mean and s.d. of three measurements. Sample fine structure, as displayed in the insets of Fig. 2e and Fig. 5, was imaged with the same confocal microscope used for actuation, but in Airyscan mode. Microparticles and aggregates can be detected due to the weak fluorescence of the azopolymer upon green light irradiation. The largest aggregates' size was estimated based on binary images obtained through local mean filtering (~5% of image size), after smoothening with a median filter accompanied by multiple erode/dilate operations in ImageJ. Example images from areas of comparable size to the actuation zones, together with the obtained particle outlines and histograms are displayed in Supplementary Fig. 9b.

## 3D actuation

Recordings of the actuation were taken by means of a CMOS camera (iDS UI-1540LE-M-GL) whose upper edge was rotated slightly towards the sample (side-top-view). The membranes were placed parallel to the measurement table and irradiated with normal laser beam incidence from the top ($\lambda = 532$ nm, $I = 1$ W·cm$^{-2}$). The Polarization state of the laser was controlled by means of a linear polarizer followed by a half-wave plate adjusting the polarization direction and, when required, by an additional quarter-wave plate to obtain circular polarization. The half-wave plate was inserted into a motorized rotating stage (Thorlabs K10CR1/M). The image collection path further contained a 40 mm achromatic lens, a 550 nm long pass filter blocking the laser beam, and an iris placed in the focal plane of the lens to increase the depth of field of the imaging system.

## Finite element model

A two-dimensional finite element model of a representative unit cell of composite with unit edge length was set up under plane-stress conditions for a mechanical analysis in the commercial software Abaqus/Standard. Periodic displacement boundary conditions were enforced on opposing edges of the simulation domain. The azopolymer was modeled with $N = 25$ non-intersecting, randomly placed and initially spherical domains, taking up 33% of the simulation area, corresponding to the azopolymer content in the physical samples. To model the irradiation-induced eigenstrain of the azo-domains, which drives the deformation of the composite, a homogeneous deformation gradient

was prescribed in the azopolymer phase. For linear and circular polarization, this deformation gradient is denoted by $\mathbf{F^{lin}}$ and $\mathbf{F^{circ}}$, and defined as follows:

$$\mathbf{F^{lin}} = \begin{pmatrix} a & 0 & 0 \\ 0 & 1/\sqrt{a} & 0 \\ 0 & 0 & 1/\sqrt{a} \end{pmatrix}, \mathbf{F^{circ}} = \begin{pmatrix} b & 0 & 0 \\ 0 & b & 0 \\ 0 & 0 & 1/b^2 \end{pmatrix} \quad (5)$$

where the stretching factors $a$ and $b$ can be adjusted and only the four left top elements corresponding to the x-y subspace are applied explicitly to the finite element model. To model the material behaviour of the SEBS matrix, the hyperelastic Marlow model[64] was fit to the tensile test data given by Kollosche et al[53] to accurately model the non-linear elastic response. Further, SEBS was considered incompressible[46], with the Poisson ratio reported in the literature being 0.49[65]. The azopolymer inclusions, which exhibit significantly higher stiffness, are modelled hypoelastically (Young's modulus $E = 4$ GPa, Poisson's ratio $v = 0.5$). They were considered incompressible, i.e., conserving their volume from one state to another in the simulation, based on previous reports on the photo-deformation of such polymers[10,50]. Their incompressibility was also considered unaffected by the surrounding SEBS matrix whose Young's modulus $E \approx 244$ kPa is orders of magnitudes lower[53]. Both phases were assumed to be isotropic. The macroscopic strains were derived from the relative displacement between the reference nodes used in the periodic boundary condition definition to represent the average position of one edge of the simulation domain each. The resulting macroscopic strains are averaged by taking the mean over a set of 9 individual simulations with different randomly initialized azo-domain distributions. Associated error intervals refer to the sample standard deviation of that set. Finally, the scale invariance of the model was verified by varying the number of azo-domains as $N = 15$, $N = 25$ and $N = 35$ at a constant 33% azo-phase for linear stretching with a = 1.125. It was observed that the variation of the output is on the order of the sample standard deviation obtained when varying the random azo-domain distributions at $N = 25$.

## Data availability

The authors declare that raw data supporting the analysis of thin film deformations is publicly available on figshare repository at https://doi.org/10.6084/m9.figshare.24123156. All other data is available from the authors upon request.

## Code availability

MatLab codes for extracting strain parameters from the raw dataset are available on figshare repository at https://doi.org/10.6084/m9.figshare.24123156.

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

## Acknowledgements

D.U. acknowledges the Norwegian University of Science and Technology for funding the PhD fellowship under project number 989454111 and The Research Council of Norway is acknowledged for the support to the Norwegian Micro- and Nano-Fabrication Facility, NorFab, project number 295864. E.D. and N.M. acknowledge the funding received by Italian Ministero dell'Università e della Ricerca under the Dipartimento di Eccellenza 2018–2022 program. We thank Hexpol A/B for kindly providing a badge of SEBS Mediprene 500120 M free of charge. Further, we acknowledge NTNU's Chemistry Department and in particular Bicheng Gao for their kind assistance in determining the molecular weight of the polymer. The results presented are part of a project that has received funding from the European Research Council (ERC) under the European Union's Horizon 2020 research and innovation programme (Grant agreement No. 949807) with starting date May 2021.

## Author contributions

E.D. and D.U. conceived and planned the work. D.U. and N.M. performed sample preparation, measurements, and data analysis for physical samples. C.H.W., J.T. and D.U. developed the mechanical modelling, performed the simulations, and analyzed simulation data. D.R.H contributed to data analysis and provided background laboratory capabilities. All authors contributed to the manuscript writing and revision.

## Competing interests

The authors declare no competing interests.
