## [Peer review file · Nature Communications]

REVIEWER COMMENTS

Reviewer #1 (Remarks to the Author):

In this manuscript, the authors reported a photo-responsive reversible shape switching behavior (in response to linearly polarized light) in an azo-SEBS composite system. The rational design and precise control are well demonstrated. Furthermore, the proposed application in free-standing 3D actuators is impressive. This work provides some new viewpoints to control the deformation direction and reversibility in light-responsive soft actuators. I recommend the acceptance of the manuscript after the following points have been well addressed.

1. The molecular structure of azopolymer has great influence on its photo-responsive behavior. I suggest that the authors provide the molecular structure, molecular weight, and UV-vis spectrum of the as-used azopolymer.
2. The authors attribute the failure of surface wrinkling on azo-SEBS composite film/PDMS to the adhesion of film and PDMS substrate (Page 6, line 11-12). However, the film/substrate properties (size, thickness, modulus and surface structure), the illumination area, light intensity and exposure time will influence its wrinkling. And oriented/disordered wrinkling has been reported on an azopolymer film/soft PDMS system under/after a polarized light illumination due to the photo-orientation/photothermal effect. So, what's the modulus of PDMS in this article? What's the size of this sample? And if decreasing the thickness of azo-SEBS composite film (without micropillar structure) and increasing the irradiation time and illumination area (i.e., blanket exposure), wrinkling will occur under/after the illumination?
3. The light intensity used in this work is relatively high, and thus the photo-thermal effect may be not ignored. (a) What's the surface temperature variation of the sample under illumination? In this case, photo thermal-induced isotropic expansion strain may influence the photo orientation-induced anisotropic strain. Therefore, will the reversible shape switching be restrained at a higher light intensity? (b) The plastic deformation of azo-SEBS composite film depends on the size of embedded azopolymer aggregates, and the larger aggregates will induce higher local heating and relax the elastic constraints in the surrounding matrix. Please give the detailed temperature values of azo-SEBS composite film with different size of embedded azopolymer aggregates.
4. The work in Ref. 20 is also related to the shape switching of the similar azo-SEBS material. What's the difference between the current study and the work in Ref. 20?
5. The authors mentioned in the article that "incorporating a mechanism for returning towards the pristine state, such as heating of the azopolymer microdomains above the glass transition temperature [20], would lead to a third reversible degree of freedom" (page 22), I suggest that the authors provide the detailed experimental data.
6. In the current study, the authors focus on the shape switching behavior of the azo-SEBS composite film. How about the pure azopolymer film without SEBS under the same conditions? As one very important comparison, the control experiment should be carried out.
7. For the as-demonstrated light-triggered actuation, the dimension of the azo-SEBS composite film is relatively small. How about the influence of the film dimension on the actuation behavior?

8. There are some minor mistakes in this article, please check it. For example: (a) In the legend of Figure 5, the sequence number of the picture should be "a, b, c", not "b, c, d". (b) In page 19, line 17, "In [54], ..." is incomplete, the reference number was seen as a part of the sentence, please check it.

Reviewer #2 (Remarks to the Author):

The manuscript entitled "Polarization-driven reversible actuation in a photo-responsive polymer composite" describes composite photoactuators consisting of nano/microparticles of amorphous azopolymer embedded in an elastic matrix. The actuation of these composites might be driven by linearly/circularly polarized light causing the deformation of azo particles. Despite on proof-of-concept character of the work, the authors attempted to explain the mechanism of photo-mechanical coupling and model the macroscopic behavior of the composite. This work will undoubtedly meet the interest of a broad community especially working in the field of photoresponsive materials and soft-actuators. I can recommend this work to consider for possible publication in Nature Communications after addressing the following issues and recommendations:

- In the first section: It would be much appreciated by the readers if the statements (on how deformations occur) would be schematically illustrated.
- The second section on FE modeling disturbs the flow a lot. I would recommend that the authors try better integrating this section into the text or shift it to Supplementary materials. The current version doesn't give much for the understanding of composite behavior and lacks conclusions.
- In the next section: Adding small schemes demonstrating the deformations in Figure 4 would be helpful. Concluding remarks are also missing. Maybe at this point, the authors could share the best parameters for such composites (size and amount of azoparticles, thickness of the layer, intensity/dose of light exposure, etc.)?
- In the section "3D actuation...": Could you please elaborate more on the following points: i) why subsequent deformations aren't cumulative by using only linearly polarized light? ii) Not clear here how the circularly polarized light affects the actuation
- It would be beneficial if the authors could measure the mechanical stresses/forces generated in the composite upon exposure to light of different polarizing.
- Absorbance spectra of the azopolymers confirming the choice of wavelength should be provided

Reviewer #3 (Remarks to the Author):

The authors describe a responsive actuator based on azobenzene-polymer micro-particles dispersed in a rubber matrix. By actuating the particles by light the rubber deforms by the directional dimension changes of the azopolymer. Because of the dichroic nature of azobenzene the deformation can be steered by the polarization direction of the actuating light source. For their study the authors deposited the azopolymer-rubber composite on PDMS of which the in-plane deformation is studied by the displacement of surface markers. The authors paid special attention to the reversibility of the effects. In the second part of the paper the authors studied free-standing films of the azopolymer-SEBS composite.

The paper is strong on the mechanical part but rather weak on the chemistry part, although the leading principle is a photochemical reaction. So after having read the paper, I am still uncertain on what mechanism the authors propose. Are they reorientating the azobenzene groups through a continuous trans-cis isomerization by which we ultimately end in an oriented trans state of the azobenzenes with their transition moment perpendicular to the polarization of the light source. Simple (polarized)FTIR would have answered these questions. Maybe full re-alignment of the azobenzene is unlikely because of the high glass transition of the azobenzene polymer. Alternatively, is it the trans-to-cis conversion which provides the dimensional change? Also, in this case one must overcome the high T_g of the polymer to make it result in larger dimensional changes, but photo-induced heating

might help (could the authors measure the temperature?). In the discussion the authors speak of the orientation of the main chain which seems to be unlikely for these acrylate main chains. More likely is it that the azobenzene moieties become oriented. The extinction coefficient of the azopolymer is high. The question is how much light reaches the bottom of the film? Do we have a gradient in time, as the azobenzene is photo-bleaching during its trans to cis conversion? Questions like this earn some attention to understand the principle better.

The methods to produce the composites and the deforming surfaces is easy and straight forward based on solution processes. A point of interest here is to what extent the sample preparation method affects the morphology of both the SEBS rubber and the azobenzene polymer particle. Because of PDMS swelling in toluene, and the in-plane stresses of coming from the evaporation process of the toluene might induce in-plane alignment which has its influence on the deformation characteristics of the particle and the SEBS. What I also miss here is a better characterization of the azobenzene particles. They are the result of phase separation as the toluene evaporates. Particle size is likely to be dependent of the evaporation rate of the toluene. Is this homogeneous over the sample and sample thickness? What are their sizes and their size distributions?

I appreciated the part of the manuscript where the authors characterize the formation of in-plane stress-induced deformation of the composite adhered to a PDMS substrate although I would have liked to see the comparison with a more rigid and good adhering glass substrate without the compliance of the PDMS. The second part of the manuscript where the authors studied free-standing composite films of azobenzene polymer and SEBS is more at 'me-too' level given the many publications of comparable phenomena seen for azobenzene modified samples. From this point of view, and even though the authors propose a very easy method to produce the films, I put some questions whether the manuscript is at the level of Nature Communications.

Reviewer #1 (Remarks to the Author):

In this manuscript, the authors reported a photo-responsive reversible shape switching behavior (in response to linearly polarized light) in an azo-SEBS composite system. The rational design and precise control are well demonstrated. Furthermore, the proposed application in free-standing 3D actuators is impressive. This work provides some new viewpoints to control the deformation direction and reversibility in light responsive soft actuators. I recommend the acceptance of the manuscript after the following points have been well addressed.

1. The molecular structure of azopolymer has great influence on its photo-responsive behavior. I suggest that the authors provide the molecular structure, molecular weight, and UV-vis spectrum of the as-used azopolymer.

We agree with the reviewer. All requested data has been included into the supplementary information under "Supplementary Figure S1" which is referred to when introducing the polymer. We point out that the absorption spectrum shown mainly refers to the trans isomeric state of azobenzene units.

2. The authors attribute the failure of surface wrinkling on azo-SEBS composite film/PDMS to the adhesion of film and PDMS substrate (Page 6, line 11-12). However, the film/substrate properties (size, thickness, modulus and surface structure), the illumination area, light intensity and exposure time will influence its wrinkling. And oriented/disordered wrinkling has been reported on an azopolymer film/soft PDMS system under/after a polarized light illumination due to the photo-orientation/photo-thermal effect. So, what's the modulus of PDMS in this article? What's the size of this sample? And if decreasing the thickness of azo-SEBS composite film (without micropillar structure) and increasing the irradiation time and illumination area (i.e., blanket exposure), wrinkling will occur under/after the illumination?

For all planar deformation measurements in this work PDMS Sylgard 184 provided by Sigma Aldrich was used as a substrate, using the recommended PDMS to cross-link agent ratio of 10:1 and a curing temperature of 65° C. The hardness is indicated by the manufacturer through a Shore durometer value of 43 (A) which, using the approximation proposed by A. N. Gent (*Gent, A. N. Rubber Chemistry and Technology 1958, 31, 896–906*) roughly translates to a Young's modulus of ~1.9 MPa, whilst ~2.6 MPa have been reported experimentally for the same product and preparation conditions (*Wang, Z. et al., Journal of Appl. Polym. Sci. 2014, 131*). This is much stiffer than the SEBS formulation employed for the composite's elastic matrix, whose Shore durometer value is indicated as 12 (A) by the manufacturer and whose elastic modulus has been reported experimentally as 244 kPa (*Kollosche, M. et al., in (ed. Bar-Cohen, Y.) 76422X, 2010*).

The size of all samples employed in this work is of several millimeters, however the illuminated areas for planar measurements are in the range (33.8 x 33.8 – 101.4 x 101.4 μm^2), see "Methods->Planar Substrate Actuation". Decreasing the thickness of these composites further in a homogeneous way would be a challenging task, however obtaining wrinkles was attempted modifying the two other factors mentioned. In detail, sample 2 (thickness $5.4 \pm 0.8 \mu\text{m}$) was illuminated on areas up to $319.5 \times 319.5 \mu\text{m}^2$, corresponding to ~100x the area used for analyzing its deformation in this work, and with irradiation doses up to ~20x the largest step dose values employed. Nevertheless, no wrinkling became evident (see Figures b,c below).

To verify the absence of wrinkles more thoroughly, we performed an additional image treatment protocol (described below) which stems from earlier attempts to characterize composite film deformations, and which was found to be particularly sensitive to wrinkling. In fact, using this approach on the deformation sequence of the suspended membrane shown in Supplementary Movie M2, wrinkles can already be

revealed after the very first illumination step (frame 2) as seen in Figure a below, whilst the wrinkles shown in Supplementary Figure S3 are directly appreciable on the movie frames, but after many more illuminations (frame 58).

Deformations on the suspended membrane shown in Supplementary Movie M2. Alternating linear horizontal and circular polarizations are used for the round (~200 μm) laser spot. (i) Raw images (after linear polarization steps): out-of-plane bulging and wrinkling can be faintly seen to appear on the later frames. (ii) Differential image qualitatively illustrating the displacement field of the raw image in the same column with respect to the first (reference) image. Displacement occurs along both x- and y-directions. (iii) Differential image obtained when replacing the reference image with its artificially deformed version, compensating the deformation by the best fitted affine transformation mapping the reference image to the frame in question. Wrinkles can already be clearly seen after the first 15 s illumination (frame 2). On later frames, bulging of the membrane becomes the dominant difference which cannot be represented/compensated by the affine transformation. Same contrast settings as in (ii). (iv) Enhanced contrast version of (iii) for easier visualization.

Also when using this method, no wrinkles were detected on the azo-SEBS film attached to PDMS (see Figures b,c below). We speculate that this might be due to the azo-SEBS composite being much softer than the PDMS, as detailed above, and to the deformation seemingly saturating at some point during the exposure (a tendency that can be appreciated already on Figure 4 in the manuscript), possibly due to mechanical constraints. Finally, we do not exclude the possibility that wrinkles could be found in the framework of a more focused investigation. The sentence on page 6 was therefore modified to “[...] mitigates effects such as gradual out of plane bulging and wrinkling.”

Same image treatment protocol as in Figure a, performed on deformation images obtained from continuous horizontally polarized illumination of different areas on Sample 2. No wrinkling becomes apparent, also when considering row (iv), which means the deformation can be reasonably represented by homogeneous scaling. The deformation also appears to saturate. Due to the larger area, longer illumination times were needed on the bottom series (c), causing the sample to drift during acquisition. This leads to drifting of the dark region (now indicating the zone where scaling displacement is compensated by sample drift). It should be noted that the deformation sequence and the optical setup in Figure (a) are different from the ones in (b,c) and comparison is mainly drawn to illustrate the sensitivity of the proposed image treatment protocol to accentuate wrinkles when present.

Protocol:

1) The image after illumination (deformation image) is compared to the reference image by means of subtracting the latter from the former (rescaling both images independently and keeping the absolute values of the resulting image matrix). This gives rise to an image qualitatively illustrating the absolute value of the displacement locally (with a dark region for the parts of the sample that are not displaced). A similar differential image was also used in Supplementary Figure S2.

2) The best affine transformation mapping the reference to the deformed image is then found using the MatLab function "imregtform". This affine transformation is applied to the reference image in order to create an artificial (deformed) image.

3) Step 1 is then applied again, this time subtracting the artificial image from the deformation image. Comparing this differential image to the one obtained in step 1 (at identical contrasts) provides a visual indication of the descriptive value of the affine transformation obtained. However, we observe that wrinkling is also automatically highlighted in this display, since the algorithm is working with a single scaling factor along any principal axis and therefore is unable to accommodate the perturbation to the (otherwise linearly increasing) displacement field caused by the wrinkles.

3. The light intensity used in this work is relatively high, and thus the photo-thermal effect may be not ignored. (a) What's the surface temperature variation of the sample under illumination? In this case, photo thermal-induced isotropic expansion strain may influence the photo orientation-induced anisotropic strain. Therefore, will the reversible shape switching be restrained at a higher light intensity? (b) The plastic deformation of azo-SEBS composite film depends on the size of embedded azopolymer aggregates, and the larger aggregates will induce higher local heating and relax the elastic constraints in the surrounding matrix. Please give the detailed temperature values of azo-SEBS composite film with different size of embedded azopolymer aggregates.

The reviewer is raising an important point here. In order to address point (a), we performed temperature measurements on azo-SEBS films on PDMS slabs under different illumination power densities. Assessing the temperature on the sample surface might be not an easy task, due to the insulating nature of the azo-SEBS films and the substrates. We conducted measurements by means of a calibrated IR-sensitive thermo-camera which has been successfully used in a previous work (A. Gliozzi *et al.*, *Nat. Commun.* 2020, 11, 2576). Initially, the sample was placed on an inverted microscope and illuminated with a focused laser beam (wavelength 532 nm, NA=0.2). We intended to measure temperature on the film surface at different positions of the illuminating spot, in particular on top of big pDR1m-co-mma aggregates and between them, as shown in Figures 5b,c. However, at the laser powers reported in the work, we failed to stably detect any significant temperature variation. This could be probably due to a too small heated volume (μm^3) and a limited spatial resolution of the thermo-imaging system. Therefore, we mounted the sample onto a separate setup, wherein the illumination laser was focused down to a 1.5 mm diameter spot. In such a configuration, we could measure the laser power-dependent temperature excess on sample type 2 and 4, as shown below.

Top line: measured temperature excess on azo-SEBS films on PDMS slabs. Sample type 2 and 4 are considered, wherein average azopolymer aggregate size and film thickness are smaller for sample 2 than sample 4. Bottom line: illustrative thermocamera images referred to sample 4 mounted on a 2-axis translational stage and being irradiated with a 532 nm wavelength CW laser at different power densities.

In a previously published work, a temperature excess of about 50° K is reported for a 100% pure DR1A film irradiated by a 488 nm wavelength CW laser at 1 W/cm² power (K. G. Yager, C. J. Barrett, "Temperature modeling of laser-irradiated azo-polymer thin films", *J. Chem. Phys.* 2004; 120: 1089–1096). Provided that the absorption spectrum of DR1 drops by about 40% when moving from 488 nm wavelength (close to the absorption peak) to 532 nm and the content ratio of pDR1m-co-mma:SEBS is 1:2, we found a substantial agreement of our measurements with the results reported in the article above.

To answer the question "will the reversible shape switching be restrained at a higher light intensity?", we invoke the plot in Figure S7, where asymptotic values of δA are plotted against the radiation doses, at different laser power regimes. During the scanning illumination at fixed laser power, the pixel dwell time (i.e. the time each point is exposed to the laser) remains constant and the different doses are provided by cumulating multiple exposures in sequential scans of the whole area. Given this, it is likely that a higher temperature is induced at a higher laser power, during the same pixel dwell time. However, plots in Figure 4 and Figure S7 do not show any significant variation in terms of asymptotic elongations and area conservation (δA) for the different laser power used. It seems thus plausible to conclude that, within the power ranges used in our measurements, a higher laser intensity does not affect significantly the deformation reversibility in samples characterized by small azopolymer aggregates.

Regarding point (b), we can first observe that samples considered here do have different azo-domain size distributions, but also different thicknesses, as described in the manuscript and summarized in Table 1. Thickness and aggregates size result both in a general increase of absorbed radiation power and thus heating, as shown by the two curves in the figure above. However, we speculate that the different temperature excess measured on sample 2 and 4 is mainly due to the different film thickness rather than a different aggregation state of pDR1m-co-mma domains. In fact, the relatively large size of the laser spot allows us to assume that the overall amount of absorbing molecules remains the same in both samples, no matter what the azopolymer distribution inside the irradiated volume is (recall that the content ratio of pDR1m-co-mma:SEBS is 1:2 for all sample types). The situation is completely different when we evaluate the deformation behaviour of sample 4 after localized illumination on top of big azopolymer aggregates or between them (see Figure 5b,c). Here, the number of absorbing units within the irradiated areas is certainly not constant. As the reviewer correctly states, big azopolymer aggregates are associated to larger

absorption and thus heating. However, a reproducible and stable measurement of locally induced temperature excess on such a small scale remains too challenging for our experimental apparatus. We believe that a dedicated setup must be built specifically for this purpose.

Finally, coming to 3D actuators, photothermal effects negatively affecting actuation are presumably related to power absorption due to the membrane thickness rather than the aggregation state of the azopolymer. Also in this case, the illumination spot is large enough (basically, an expanded, collimated laser beam) to average/integrate the concentration distribution of pDR1m-co-mma in SEBS.

In the “Discussion” section, we emended our speculation on the role of temperature in deformation mechanisms by adding explicit reference to the figure above (included into the Supplementary Materials as Figure S14).

4. The work in Ref. 20 is also related to the shape switching of the similar azo-SEBS material. What’s the difference between the current study and the work in Ref. 20?

The referenced article by Ryabchun and Bobrovsky represents a very important piece of work that is propaedeutic to our study. In that work, micrometric domains of Liquid Crystal copolymer PAAzo bearing mesogenic and azobenzene photoactive groups are studied. Optical (birefringence) and opto-mechanical effects are obtained by exploiting the photochromism of azobenzene and the capabilities of self-reorganization of the Liquid Crystalline phase. In our work, no LC compounds are involved and the local deformation of azopolymer domains are related to a polarization-driven phenomenon sometimes referred to as directional photofluidization. Very briefly, we can highlight the following novel content in our manuscript in the following points:

- we provide a method for embedding azopolymer domains within SEBS at higher density;
- we provide a detailed and quantitative method to analyse and evaluate the extent of deformation reversibility induced in the composite material (on a larger scale than the single azo-domain) by polarization-driven effects. We find that dense and sub-micrometric aggregations of pDR1m-co-mma within the elastomeric matrix do play a significant role in facilitating reversibility;
- we provide two examples of complex actuation in macroscopic membranes where the polarization is exploited as a control mean. We put in evidence a direct relationship between polarization state of the illuminating radiation and actuation movement produced on the macroscopic light-responsive film.

5. The authors mentioned in the article that “incorporating a mechanism for returning towards the pristine state, such as heating of the azopolymer microdomains above the glass transition temperature [20], would lead to a third reversible degree of freedom” (page 22), I suggest that the authors provide the detailed experimental data.

We thank the reviewer for highlighting a statement that we acknowledge could be confusing. In our work we could show that a switching regime can be obtained for azo-domains embedded in SEBS, possibly transitioning between prolate and oblate ellipsoidal shapes upon illumination with linear or circularly polarized light. Assuming an initial spherical shape of the domains, we show that such a pristine status is unlikely to be recovered by employing polarization-driven effects only. Instead, in [20] (now [21], with the updated reference list), Ryabchun and Bobrovsky suggest a shape recovery approach based on exposure to high temperature or any other means producing an isotropic softening of the polymer (see pag.4 in [21]). The purpose of our statement is to include such an option to accomplish a full reversibility of the film deformation also on the macroscopic scale.

We rephrase the statement in the following way: “In order to recover the very initial microdomain state, mechanisms based on heating the azopolymer above the glass transition temperature have been proposed and experimentally validated [21].”

6. In the current study, the authors focus on the shape switching behavior of the azo- SEBS composite film. How about the pure azopolymer film without SEBS under the same conditions? As one very important comparison, the control experiment should be carried out.

We thank the reviewer for this excellent suggestion. A pure pillar-patterned azopolymer film was deposited on PDMS as usual and measurement results are included in Supplementary Figure S5. In particular, the relative area expansion per cycle is seen to not decay, but stay constant, at a similar value to ϵ_1 . This is what would be expected for sequentially applying the tensors F^1 and F^2 of Equation (2), where $\epsilon_1 = \delta$ for horizontal stretching (i.e., applying F^1) and $\delta A_{\text{cycle}} = -(-1/2 \delta - 1/2 \delta) = \delta$ theoretically. We also observe that $-\epsilon_2/\epsilon_1 \approx 0.5$ for all cycles, indicating that F^1 keeps being applied cumulatively for each horizontally polarized illumination. This contrasts with the switching behavior of the composite characterized by a decay of δA_{cycle} and the emergence of a regime where $-\epsilon_2/\epsilon_1 \approx 1$ (area conserved also for the horizontal polarization step). The above observations and Supplementary Figure S5 are referred to in the end of section 1, when discussing the comparison with the usual cumulative deformation behavior of pure azopolymers. Supplementary Figure S5 is also referred to in Section 3, pointing to the similarity between these measurements and the measurements taken on top of a single large azopolymer aggregate of Sample 4, which are in good agreement. Finally, it should be noted that the graph displays only the first 15 illumination cycles, due to the cumulative deformations degrading the pillar lattice and rendering the fitting more difficult for the last cycles. All 20 cycles are displayed in Supplementary Movie M3.

7. For the as-demonstrated light-triggered actuation, the dimension of the azo-SEBS composite film is relatively small. How about the influence of the film dimension on the actuation behaviour?

We share the reviewer’s curiosity in wondering how the size and (more generally) the geometry of the actuator play a role in the actuation mechanism. The question is indeed very broad and would require dedicated efforts to conduct a systematic study. However, within the scope of the present work, we performed additional observations as described below.

First, a circular membrane with similar radius and larger thickness as compared to the actuator in the manuscript (thickness about 40 μm) is considered. Generally, the directional actuation mechanism is occurring as reported. However, we observe a smaller bending amplitude and a slower response time, given similar illumination conditions (Movie R1, for review only).

Then, the case of a large cantilever membrane is considered (see Movie R2, for review only). Here, the surface area is about twice as large as in the trapezoidal cantilever shown in Supplementary Movie M6, while the thickness is the same. Beside similar macroscopic actuation movements as in Movie M6, we notice a longer transition time to the new conformational state, when the illumination polarization is varied. Some wavy behaviour of the membrane can be observed while the system is reaching a stable configuration. This is possibly related to the accommodation all local deformations triggered by light.

The two examples above seem to indicate that a favourable range of surface to thickness ratio exists for the actuators to simultaneously display ample and smooth bending behaviour. Investigations on this subject are still ongoing and would require a careful modelling of the photo-mechanical interaction of azopolymer units with light, including the role of the mechanical constraints.

8. There are some minor mistakes in this article, please check it. For example: (a) In the legend of Figure 5, the sequence number of the picture should be “a, b, c”, not “b, c, d”. (b) In page 19, line 17, “In [54], ...” is incomplete, the reference number was seen as a part of the sentence, please check it.

We thank the reviewer for the careful reading. We corrected the typos.

Reviewer #2 (Remarks to the Author):

The manuscript entitled “Polarization-driven reversible actuation in a photo-responsive polymer composite” describes composite photoactuators consisting of nano/microparticles of amorphous azopolymer embedded in an elastic matrix. The actuation of these composites might be driven by linearly/circularly polarized light causing the deformation of azo particles. Despite on proof-of-concept character of the work, the authors attempted to explain the mechanism of photo-mechanical coupling and model the macroscopic behavior of the composite. This work will undoubtedly meet the interest of a broad community especially working in the field of photoresponsive materials and soft-actuators. I can recommend this work to consider for possible publication in Nature Communications after addressing the following issues and recommendations:

1. In the first section: It would be much appreciated by the readers if the statements (on how deformations occur) would be schematically illustrated.

A sketch showing how light-driven deformations affect the topography of the patterned films has been added in the “Methods” section (Figure 8), where the analysis method is described. In particular, the case of engineering strains along two orthogonal axis resulting from a linearly-polarized illumination is considered as an illustrative example. For sake of generality, the linear polarization is oriented with an angle θ with respect to the x-axis. Engineering strains are explicitly indicated in the direct space. Furthermore, a pictorial view of the corresponding deformation of the hexagonal cell in the Fourier space is also presented, according to the k-vector transformation embodied in Equation (1) in the manuscript and used for fitting experimental data.

2. The second section on FE modelling disturbs the flow a lot. I would recommend that the authors try better integrating this section into the text or shift it to Supplementary materials. The current version doesn't give much for the understanding of composite behaviour and lacks conclusions.

We thank the reviewer for this comment. In the “FE model” section, we intend to justify a key assumption underlying parts of the analysis conducted on azo-SEBS film deformations over areas much larger than the single domain size: that azo-domain eigenstrains translates linearly into macroscopic strains of the composite. This conclusion of our FE study brings an important insight into the composite behaviour that is particularly useful for the interpretation of results. We demonstrate its validity within the range of strain amplitudes achieved experimentally, so that we can safely infer shape transition mechanisms for individual azo-domains by studying macroscopic deformations induced by light. Therefore, we believe this section has a significant value in the frame of this article and should be presented after the phenomenological description of the observed effects as it both facilitates the understanding of the mechanical behaviour of the composite and outlines a practicable and viable modelling approach for the current as well as future investigation. However, in an attempt to accomplish the reviewer request, we specify the rationale behind the FE modelling section by adding an explanatory text recalling its relatedness to the previous section and highlighting the key conclusion.

3. In the next section: Adding small schemes demonstrating the deformations in Figure 4 would be helpful. Concluding remarks are also missing. Maybe at this point, the authors could share the best parameters for such composites (size and amount of azoparticles, thickness of the layer, intensity/dose of light exposure, etc.)?

This comment is partially addressed in the discussion of the previous point (1). We believe the new Figure 8 in the “Methods” section can facilitate the interpretation of Figures 4,5 in the manuscript. In addition, concluding remarks on the deformation of patterned films have been added before discussing free-standing actuators, as suggested by the reviewer.

4. In the section “3D actuation...”: Could you please elaborate more on the following points: i) why subsequent deformations aren’t cumulative by using only linearly polarized light? ii) Not clear here how the circularly polarized light affects the actuation

We thank the reviewer for raising such important points. In order to provide a satisfactory answer to (i), we conducted new measurements on a sample of type 1. In particular, we modified the illumination protocol and employed a sequence of linearly polarized states (alternating X-POL and Y-POL) instead of the sequence LIN-POL:CIRC-POL described in the manuscript, then we applied the same analysis process. The resulting graph has been included in Supplementary Figure S12 and is referred to at the beginning of section 3. The reversibility features discussed so far in the manuscript, namely a significant drop in the relative area expansion δA_{cycle} and a ratio of $-\varepsilon_2/\varepsilon_1 \approx 1$ during horizontally polarized illumination steps, are clearly present also in this case. For a comparison with the non-reversible case of pure azopolymer (where $\delta A_{\text{cycle}} \sim \varepsilon_1$ and $-\varepsilon_2/\varepsilon_1 \approx 0.5$), one may consider the measurements on single large azopolymer aggregates (manuscript section 3, Figure 5b) or the measurements provided following the comment point (6) by Reviewer 1 concerning measurements on pure azopolymer films (Supplementary Figure S5a), where this was not the case. Further examples of cumulative deformation effects in pure azopolymer systems are also provided in the end of section 1 of the manuscript.

A direct-space video of the X-POL:Y-POL illumination protocol is provided as Movie R3 (for review only). It was not included as a further supplementary movie due to most likely adding little value by eye. In fact, the difference between this sequence and the sequence shown in Supplementary Movie M1 (using the X-POL:CIRC-POL protocol), is rather subtle, in that the sample is reversibly switching between an x-axis stretched and another (either in-plane expanded or vertically stretched state) in both cases. If of interest to the reviewer, the best way we found to highlight a difference between both sequences in direct space by eye was to loop frame 1 (before exposure) and frame 3 (after the end of the first illumination cycle) in both cases side-by-side, which is shown in Movie R4 (for review only). There, the conventional sequence (left) shows in-plane expansion with respect to the initial state before any exposure, whilst the new sequence shows vertical stretching with respect to the initial state. In both Movie R3 and Supplementary movie M1, the substrate then seems to switch between this state and the horizontally stretched state induced by horizontal polarization.

In conclusion, this further experiment allowed to show that the switching behavior observed on planar substrates is not limited to the LIN-POL:CIRC-POL sequence, but is also present when employing sequences of different linear polarizations, reinforcing the observations made on the 3D circular membrane actuator, when employing sequential of linear polarizations only.

Coming to the reason why this happens, we can invoke the role of the SEBS elastomeric matrix and its interface with the azopolymer particles, as briefly addressed in the beginning of the Discussion section. Nevertheless, whilst the experimental observations are quite clear on the existence of an emerging switching behaviour and its opposition to observations both in literature and concerning the control experiments in this work, we acknowledge that a precise description of the interaction mechanism with the elastomeric matrix will require further efforts (for example employing azopolymers with different glass transition temperatures as host particles etc.).

Concerning point (ii), we conducted additional observations on cantilevered membranes, where a circular polarization is also used for illumination. In Movie R5, two co-planar cantilevers with similar widths and different lengths are illuminated with a linear polarization transverse to the cantilever axis, so that they are both brought in a position as represented in Figure 7e in the manuscript. In this discussion, for sake of simplicity, we might refer to this state as “longitudinal bending”. Then, the linear polarization is rotated, causing a corner bending similar to Figure 7c,d. The polarization is finally varied to circular. Both cantilevers revert bending, quickly reaching a planar configuration recalling the very initial one. However, as we continue illuminating the sample, some smooth convexity is observed to appear. According to our finding, this effect results from the adirectional in-plane expansion of azopolymer domains on top of the membranes. In the figure below (also inserted in the Supplementary Material as Figure S13), some representative frames of Movie R5 are illustrated, wherein the initial and intermediate states of the

membranes can be more easily appreciated. Reference to Figure S13 has been added within the “3D actuation of free membranes” section when Figure 7 is discussed.

5. It would be beneficial if the authors could measure the mechanical stresses/forces generated in the composite upon exposure to light of different polarizing.

We agree on the value and usefulness of such an information. However, available DMA systems for dynamo-mechanical characterizations require sample properties that are beyond the capabilities of our fabrication method. In particular, the ratio thickness-area of our actuator self-standing membranes is too small for a proper mounting into conventional equipment and the films are comparably soft ($E \sim 244$ kPa for SEBS). As an alternative, azo-SEBS films on PDMS slabs could be employed, but the presence of the PDMS substrate (elastic modulus roughly one order of magnitude larger than SEBS) would hinder the measurement of the light-responsive force/stress on the active film surface. Finally, we observe that in conventional DMA systems, measurements of local strain/stress can be hardly performed as average/integral characterization is provided instead.

We note that, while commercial equipment seems inappropriate, such measurements may be within reach in the future and would be interesting to explore. For example, measurements of active light-induced tension in LCN films of comparable dimension and with slightly more favourable mechanical properties ($E > 1$ MPa, chemically cross-linked) were reported in (Donato, S. et al. *Macromolecular Rapid Communications* 2023, 44, 2200958). The comparison of mechanical properties and generated active tension being a major part of the latter study, the authors employed a custom-made force measurement set-up (see e.g., Figure 7A of the cited study), whose implementation however lies outside of the scope of this work.

Finally, however, a very rough estimate of the generated stresses can of course be obtained using the elastic modulus of SEBS (~ 244 kPa) and the observed maximal strains of $\sim 1\%$, leading to an estimated maximal stress of ~ 2.4 kPa. In this context, we point out that a side-result of the mechanical model in section 2 was that the SEBS matrix is accommodating the inclusions' deformation via lower local strains (than the ones imposed on the azopolymer domains) and the SEBS strain therefore is not expected to significantly surpass the $\sim 1\%$ deformation measured macroscopically for the composite.

6. Absorbance spectra of the azopolymers confirming the choice of wavelength should be provided

We added our measured absorbance spectrum of the azopolymer in Supplementary Figure S1 together with other related information.

Reviewer #3 (Remarks to the Author):

The authors describe a responsive actuator based on azobenzene-polymer micro-particles dispersed in a rubber matrix. By actuating the particles by light the rubber deforms by the directional dimension changes of the azopolymer. Because of the dichroic nature of azobenzene the deformation can be steered by the polarization direction of the actuating light source. For their study the authors deposited the azopolymer-rubber composite on PDMS of which the in-plane deformation is studied by the displacement of surface markers. The authors paid special attention to the reversibility of the effects. In the second part of the paper the authors studied free-standing films of the azopolymer-SEBS composite.

The paper is strong on the mechanical part but rather weak on the chemistry part, although the leading principle is a photochemical reaction. So after having read the paper, I am still uncertain on what mechanism the authors propose.

Are they reorientating the azobenzene groups through a continuous trans-cis isomerization by which we ultimately end in an oriented trans state of the azobenzenes with their transition moment perpendicular to the polarization of the light source. Simple (polarized)FTIR would have answered these questions.

The reviewer is raising a relevant point here. Directional deformation in side-chain azopolymers is well known and widely reported in literature for several configurations, including flat films (*Adv. Mater.* 2012, 24, 2069-2103; *Nat. Commun.* 2012, 3, 989), patterned surfaces (*ACS Appl. Polym. Mater.* 2022, 4, 7, 4993–5000; *Angew. Chem. Int. Ed.* 2014, 53, 12116-12119; *J. Phys. Photonics* 2021, 3 034013) and nanoparticles (*Chem. Commun.* 2011, 47, 4757–4759; *J. Phys. Chem. B* 2018 122, 2001-2009). Interesting reviews on this topic can be found here: *J. Polym. Sci. Part B: Polym. Phys.* 2013, 52, 163–182, and here: *Nanophotonics* 2018, 7, 1387–1422.

While there is a general agreement on the crucial role of azobenzene re-orientation upon cyclic trans-cis photoisomerization (Weigert effect, a good description can be found in Viswanathan, N. K. et al., *J. Mater. Chem.* 1999, 9, 1941–1955), the mechanism underlying viscoplastic deformations driven by polarized illumination is still intensely debated, as pointed out in a very recent authoritative review (*Angew. Chem. Int. Ed.* 2019, 58, 9712–9740). Light-induced Surface Relief Gratings (SRG) constitute probably one of the most popular workbenches used to study photo-motion in azopolymers, which often occurs at temperatures well below T_g (*J. Mater. Chem.* 1999, 9, 1941–1955, *Soft Matter*, 2016, 12, 2593-2603). In order to give an account of the many observed phenomena, several models have been proposed so far, among which we recall: pressure-gradient model (*J. Phys. Chem.* 1996, 100, 8836), optical field gradient model (*Appl. Phys. Lett.* 1998, 72, 2096; *Chem. Mater.* 2000, 12, 1585); asymmetric diffusion model (*Pure Appl. Opt.* 1998, 7, 71); photoinduced molecular diffusion model (*ACS Nano* 2009, 3, 1573); anisotropic photofluidization (*RSC Adv.* 2016, 6, 27087–27093; *Nat. Mater.* 2005, 4, 699–703). Furthermore, an interesting orientation model has been proposed for side-chain azopolymers, whereby the reorientation of azo-units causes a reorientation of the polymer backbones so that a macroscopic deformation results from the strong mechanical coupling between active azobenzene molecules and (passive) polymer network (*J. Chem. Phys.* 2011, 135, 044901; *J. Phys. Chem. Lett.* 2017, 8, 1094–1098; *Phys. Chem. B* 2019, 123, 3337–3347).

Given the framework briefly summarized here, providing new insights on fundamental aspects governing polarization-induced anisotropic deformations in azopolymers is clearly beyond the scope of the present work. However, in agreement with the reviewer, we can certainly prove that a continuous illumination of the azo-SEBS composite with a linearly polarized laser results into well-known birefringence and dichroism. This is illustrated in the figure below, showing optical images (transmission) of our azo-SEBS composite collected in cross-polarizer configuration before and after illumination with a linearly polarized laser. Local changes in transmitted intensity can be referred to corresponding changes in dipole momentum orientation as discussed above. Worth to observe that re-orientation of azobenzene dipole momenta occurs on time scales that are typically faster than morphological deformations, although both effects are mutually connected, as will be addressed in more detail under point three.

Wide-field transmission of azo-SEBS film with crossed polarizer-analyzer pair. A squared area is irradiated with a linearly polarized laser, oriented at 45 degrees with respect to the polarizer. Illumination conditions correspond to a cycle of illumination as used in Figure 2 of the manuscript (wavelength $\lambda = 561$ nm, intensity $I = 1.3 \text{ W}\cdot\text{cm}^{-2}$, step time $t = 15$ s) and show the sample (a) before illumination, (b) right after illumination, (c) after 2 minutes of waiting time and (d) after a second illumination step with circular polarization of the laser beam. Difference in the transmitted light intensity can be accounted to a relevant change of dipole momenta orientation in azo-units, thus leading to birefringence and dichroism.

Maybe full re-alignment of the azobenzene is unlikely because of the high glass transition of the azobenzene polymer. Alternatively, is it the trans-to-cis conversion which provides the dimensional change? Also, in this case one must overcome the high T_g of the polymer to make it result in larger dimensional changes, but photo-induced heating might help (could the authors measure the temperature?).

As the reviewer correctly points out, trans-to-cis conversion can lead dimensional changes in polymers containing azobenzene moieties by itself. In general, however, the effect of trans-to-cis conversion is known to lead to isotropic effects, apart from cases where the azobenzene molecules are embedded in an underlying anisotropic network (such as liquid crystal networks, LCNs), which is clearly not the case here. Isotropic volume expansion due to trans-to-cis conversion has however been shown also in poly(DR1 acrylate) (pDR1a) thin films, via measurements of the change in film thickness (*Tanchak, O. M. & Barrett, C. J. Macromolecules 2005, 38, 10566–10570*), where the authors also illuminated their samples in an integrating sphere to exclude any azobenzene moiety re-orientation effects. They pointed out that a sufficiently high content of azobenzene moieties (>35 mol% in their case) was necessary to see an appreciable effect. In this work, the polymer employed has 15 mol% according to the manufacturer and we do not see a dominant contribution of an isotropic effect, based on the measured deformation graphs.

Coming to the question on temperature measurement, we report here the temperature characterization performed on azo:SEBS samples, which has been already shown and discussed above, to address a comment by Reviewer 1.

We performed temperature measurements on azo-SEBS films on PDMS slabs under different illumination power densities. Assessing the temperature on the sample surface might be not an easy task, due to the insulating nature of the azo-SEBS films and the substrates. We conducted measurements by means of a calibrated IR-sensitive thermo-camera which has been successfully used in a previous work (A. Glozzi *et al.*, *Nat. Commun.* 2020, 11, 2576). In such a configuration, we could measure the laser power-dependent temperature excess on sample type 2 and 4, as shown below.

Top line: measured temperature excess on azo-SEBS films on PDMS slabs. Sample type 2 and 4 are considered, wherein average azopolymer aggregate size and film thickness are smaller for sample 2 than sample 4. Bottom line: illustrative thermocamera images referred to sample 4 mounted on a 2-axis translational stage and being irradiated with a 532 nm wavelength CW laser at different power densities.

As one can see, sample temperatures always stay below the azopolymer T_g , even when the highest laser power is employed for illumination. Many published works have shown that anisotropic deformation preferentially occurs below T_g (we recall here: *J. Mater. Chem.* 1999, 9, 1941–1955; *Nat. Mater.* 2005 4, 699–703; *Soft Matter*, 2016, 12, 2593–2603). At high laser power regimes (and therefore, higher temperatures), anisotropic deformations disappear, as well as the optically induced birefringence (*Materials* 2020, 13, 2464), whilst isotropic modifications become dominant.

In the discussion the authors speak of the orientation of the main chain which seems to be unlikely for these acrylate main chains. More likely is it that the azobenzene moieties become oriented.

As mentioned in a previous point, azobenzene moiety orientation is clearly present and can be shown also in our samples. However, while being linked to each other, azobenzene moiety orientation and mechanical deformation in side-chain azopolymers are not necessarily equivalent. A detailed study (*RSC Adv.* 2019, 9, 20295) on the emergence and erasure of surface relief gratings, whilst simultaneously monitoring the bulk birefringence grating (associated to the azobenzene moiety orientation) was done by Santer and co-workers using two different side-chain azopolymers of which one, pDR1a-co-mma, is very similar to the one employed in this work. In particular, bulk birefringence can emerge faster and saturate before the maximal deformation of the film surface. Also, it was shown that by shifting the interference pattern, the physical surface relief grating could be erased, all whilst inscribing a new shifted birefringence grating. In that case the authors report ending up with a flat surface, which however has an inscribed underlying

birefringence grating stemming from azobenzene orientation. This birefringence grating then decays in the dark (due to azobenzene moiety orientation relaxation decay) without altering the film surface visibly.

In this context, it is worth to mention that some authors have made observations favouring the hypothesis of azobenzene moiety's interaction with the polymer backbones driving the polarization-dependent deformation of different side-chain azopolymers. Recently, Liu et. al., through studying wrinkling phenomena in a side-chain azopolymer reported temporally separated deformation phenomena (*Angew. Chem. Int. Ed. Engl.* 2022, 61, e202203715). The authors first observed weak film expansion in the direction perpendicular, followed by strong expansion in the direction parallel to the incident polarization, which they explicitly linked to sequential orientation of first the azobenzene side-chains (in the perpendicular direction), followed by orientation of the main-chains (in the parallel direction). In an earlier and well-known work, Bublitz et al. analyzed the deformation of two different azopolymers with identical azobenzene moieties but different main-chain architectures (*Appl Phys B* 2000, 70, 863–865). The authors reported that the polymer with stiff and short-spaced main-chains elongated strongly in the direction parallel to the incident polarization (which they attributed to parallel main-chain orientation following the perpendicular azobenzene side-chain orientation), whilst the polymer with softer backbone elongated weakly in the direction perpendicular to the incident polarization (which they attributed to perpendicular azobenzene-moiety orientation alone). Finally, as noted previously, other groups have theorized the photo-orientation model for glassy side-chain azopolymers (*Phys. Chem. B* 2019, 123, 3337–3347) and for DR1 containing acrylate-backbone polymers in particular (*J. Phys. Chem. B* 2018, 122, 6, 2001–2009). Nevertheless, given the multitude of observations and mechanisms proposed by the community (as outlined in point one), these considerations are still a hypothesis, although one that seems to have gained ground in recent years. It does not seem that this work can provide decisive experimental evidence for or against any of those hypotheses. It may however be used as a starting point for studying the interactions of the photo-deformation of glassy side-chain azopolymers with external matrices, in order to better understand the photo-softening effects and the forces involved, a matter still explicitly debated in some of the works cited above. In this regard, we expect the experimental evidence of a clear interaction of the azopolymer deformation with the very soft elastomeric matrix to be of interest to the community.

The extinction coefficient of the azopolymer is high. The question is how much light reaches the bottom of the film? Do we have a gradient in time, as the azobenzene is photo-bleaching during its trans to cis conversion? Questions like this earn some attention to understand the principle better.

The extinction coefficient of the polymer is indeed high, as can be seen from the absorption spectrum included in Supplementary Figure S1, which was taken at a concentration of 0.0025 mg/mL. The estimated absorption length of the polymer from this spectrum is of several hundred nanometers which is the right order of magnitude with respect to reports (*J. Phys. Chem. B* 2003, 107, 9736–9743; *Appl. Opt. AO* 1993, 32, 7277–7280) for similar DR1 side-chain acrylate polymers (note that mol% loading of the dye will have an influence, the polymer employed here has a relatively low chromophore content of 15 mol%). For a 33 wt% fraction of azopolymer in this work the estimated average absorption length is ~1.7 μm at 561 nm. Therefore, one may expect most of the effect (on the planar substrates) to happen at the film surface, as well as one can consider this the reason for ample bending in the 3D actuator case (see manuscript Figure 1c,d).

Note that depth attained by the illumination can be slightly altered (in time) by the phenomenon of dichroism (absorption of polarized light decreasing due to re-orientation of azobenzene moieties perpendicular to the polarization direction and thus absorbing less), which was a point of study in early work concerning optical storage and use of azopolymers for holography. A short report for this effect in a pDR1a polymer can be found in (Rochon, P., et al. *Appl. Opt. AO* 1993, 32, 7277–7280). However, for comparable irradiation conditions (4 W/cm², 514.5 nm) the reported change of absorption (which will affect the above absorption length) along the polarization is reported as <20% and the timescale is fast

(~0.2 s for saturation), happening at the speed of azobenzene moiety orientation rather than the speed of photo-deformations (as briefly touched upon in the previous point).

The azopolymer employed in this work is expected to suffer from photobleaching, as well as any other polymeric compound containing azo-dyes does. Although we had not performed precise measurements of photobleaching dynamics on such a common dye as DR1, we can state that, in the illumination conditions used here for both patterned films and 3D actuators, we have no evidence of significant decrease of photo-responsivity after long time irradiations. In particular, cantilevered membranes have been actuated with CW laser irradiation for time intervals as long as 5 hours, without apparently losing actuation/deformation capabilities at any significant level.

The methods to produce the composites and the deforming surfaces is easy and straight forward based on solution processes. A point of interest here is to what extent the sample preparation method affects the morphology of both the SEBS rubber and the azobenzene polymer particle. Because of PDMS swelling in toluene, and the in-plane stresses of coming from the evaporation process of the toluene might induce in-plane alignment which has its influence on the deformation characteristics of the particle and the SEBS.

We acknowledge that the reviewer raises a valid point here. In particular, stress formation can easily be observed when preparing films of pure azopolymer pDR1m-co-mma with the casting method used in this work and manifests itself through cracks in the azopolymer film (see figure below). In addition, illuminating thin flakes of this cracked film with the confocal scanning laser was observed to induce further cracks in a “catastrophic” manner, most likely owing to sudden stress release when inducing photo-mechanical changes in the azopolymer. It seems reasonable that the azopolymer shrinking during solvent evaporation could lead to compressive stress stored in the underlying PDMS substrate, which in turn would exert an expansive force on the azopolymer flakes. We note, however, that the pure azopolymer is expected to have a Young’s modulus comparable to poly(methyl methacrylate) (PMMA), in the GPa range. The SEBS grade employed here (SEBS 500120, Hexpol A/B) is orders of magnitude lower, with the Young’s modulus reported as 244 kPa (Kolloosche, M. et al., in (ed. Bar-Cohen, Y.) 76422X (2010)). This may explain why such cracks are not observed on the azo-SEBS composite samples in this work.

Left: Cracks in a thin film of pure azopolymer pDR1m-co-mma due to stresses arising during solvent evaporation. Right: Further cracks arise in a sudden manner when continuously irradiating flakes of the cracked film with constant laser intensity.

Nevertheless, it is likely that some stress is also present in the composite samples. In lack of a direct method to assess these stresses the following observations can be reported:

- For the 3D actuator membranes, no intrinsic curvature was observed when delaminating the membranes from both PDMS slabs, something we would expect to see if stresses were vastly different on either side of the membrane (i.e., inhomogeneous throughout the thickness of the membrane).
- For the planar analysis, whilst differences with respect to the reversibility of different samples (cast at different solvent contents) were evident, the amplitude of the horizontal deformation (ϵ_1) with given illumination conditions was comparable for all samples. This would not be likely if large differences in pre-illumination stress between different samples were playing a dominant role.

Finally, and in a more general context regarding the influence of drying dynamics on the samples, we can report the following observation. The use of spacers, which had been initially sought to help control the thickness of the films, was abandoned as it tended to lead to complex drying dynamics and films that were appreciably less homogeneous by eye (see figure below). Allowing the molds to freely move closer during solvent evacuation on the other hand was observed to lead to more homogeneous and potentially less pre-stressed films.

Example samples of type 4 made from the same solution but avoiding/using spacer tape between the two PDMS layers on the left/right. Movement of the material in the case of spacer employment (right) lead to streaks and less homogeneous appearance, due to the large volume of solvent that are evacuated from the initial solution.

What I also miss here is a better characterization of the azobenzene particles. They are the result of phase separation as the toluene evaporates. Particle size is likely to be dependent of the evaporation rate of the toluene. Is this homogeneous over the sample and sample thickness? What are their sizes and their size distributions?

Investigating the particle size distribution and its homogeneity is indeed an important point when considering the effect of large particles on reversibility. To this aim, confocal images of the samples on areas comparable to the illumination areas, but without pillar lattices, were acquired and rendered binary. The particle distributions were then estimated using ImageJ, as described in “Methods→ Further Planar

Substrate Characterization". Examples are shown in Supplementary Figure S9 and the histograms have been updated to include data from 5 images for each sample, acquired on distant areas (several millimetres apart) for each sample. All images together with the detected particle outlines are shown below, together with the associated particle distribution histograms. Note that this analysis is mainly valid for detecting the distribution of the largest aggregates/particles present.

Left: Particle distribution histograms of all four samples. Right: Underlying confocal images from distant areas with detected particle outlines (white).

In the plane of the substrates, samples seem rather homogeneous and can clearly be distinguished from each other. As to the homogeneity of particle sizes in the out-of-plane direction, the same approach can hardly be applied, due to the short absorption length of the polymer and hence of the composite (several hundred nanometres), making particles buried deep less visible. However, the SEM cross-section images shown in Supplementary Figure S9 do not indicate flagrant inhomogeneities. A magnified example is provided below.

SEM cross-section image of Sample 2. Particles can be distinguished by their round shapes.

I appreciated the part of the manuscript where the authors characterize the formation of in-plane stress-induced deformation of the composite adhered to a PDMS substrate although I would have liked to see the comparison with a more rigid and good adhering glass substrate without the compliance of the PDMS.

The reviewer raises a relevant point here and casting the azo-SEBS layers directly on a glass substrate was attempted in an early stage of this work. However, adhesion on glass was observed to be less strong and the samples appeared appreciably less homogeneous also in this case. The latter may be due to, as the reviewer correctly pointed out in the above, drying not being the only mechanism evacuating solvent from the film. Instead, Toluene is also evacuated through swelling into the (thick) PDMS layers on both sides using the fabrication method applied in this work.

We would like to point out that, as already discussed previously, the Young's modulus of the SEBS grade employed here was reported as ~244 kPa (Kollosche, M. et al., in (ed. Bar-Cohen, Y.) 76422X (2010)), which is an order of magnitude lower than the Young's modulus of the PDMS formulation employed (~2 MPa, see also reply to point 2 raised by Reviewer 1). From this point of view the PDMS is of limited compliance, considering that one measures small surface deformations directly on a much softer azo-SEBS film.

The second part of the manuscript where the authors studied free-standing composite films of azobenzene polymer and SEBS is more at 'me-too' level given the many publications of comparable phenomena seen for azobenzene modified samples. From this point of view, and even though the authors propose a very easy method to produce the films, I put some questions whether the manuscript is at the level of Nature Communications.

In our opinion, the possibility of reversibly inscribing polarization-dependent shapes into these free-standing membranes, such as to cause, for example, continuously rotating features constitutes a significant novelty for light-triggered actuators, as well as strengthening the validity of the preceding analysis on thin films.

REVIEWERS' COMMENTS

Reviewer #1 (Remarks to the Author):

In the revised version, the main concerns have been well addressed. It can be considered for the acceptance of the publication.

Reviewer #2 (Remarks to the Author):

All the issues have been properly addressed by the authors. I appreciate the authors for the detailed answers and amount of additional experiments made. I can recommend the manuscript for publication in Nature Communications in its current state.

Reviewer #3 (Remarks to the Author):

The authors have responded extensively to the remarks and questions of the reviewers. Often by referring to what is known from (indeed rather extensive) literature. But in literature there is not always agreement on the exact mechanisms of, for instance, the photo-conversion of the azobenzene and its role in the photo-induced deformation processes. So one can pick the one explanation that fits best with the observations. In that sense I had hoped for a more chemical confirmation and support of the authors, but I guess they focussed more on the phenomenological effects of the composite films under exposure with light. For the remaining I am fine with their improvements and I can agree with the paper after its revision.

Reviewer #1 (Remarks to the Author):

In the revised version, the main concerns have been well addressed. It can be considered for the acceptance of the publication.

We thank the reviewer for the positive assessment.

Reviewer #2 (Remarks to the Author):

All the issues have been properly addressed by the authors. I appreciate the authors for the detailed answers and amount of additional experiments made. I can recommend the manuscript for publication in Nature Communications in its current state.

We thank the reviewer for the positive assessment.

Reviewer #3 (Remarks to the Author):

The authors have responded extensively to the remarks and questions of the reviewers. Often by referring to what is known from (indeed rather extensive) literature. But in literature there is not always agreement on the exact mechanisms of, for instance, the photo-conversion of the azobenzene and its role in the photo-induced deformation processes. So one can pick the one explanation that fits best with the observations. In that sense I had hoped for a more chemical confirmation and support of the authors, but I guess they focussed more on the phenomenological effects of the composite films under exposure with light. For the remaining I am fine with their improvements and I can agree with the paper after its revision.

We thank the reviewer for the positive assessment and the critical comment on the topic.